# Bregman Divergence for Stochastic Variance Reduction: Saddle-Point and Adversarial Prediction

**Zhan Shi**      **Xinhua Zhang**
University of Illinois at Chicago
Chicago, Illinois 60661
{zshi22,zhangx}@uic.edu

**Yaoliang Yu**
University of Waterloo
Waterloo, ON, N2L3G1
yaoliang.yu@uwaterloo.ca

## Abstract

Adversarial machines, where a learner competes against an adversary, have regained much recent interest in machine learning. They are naturally in the form of saddle-point optimization, often with *separable* structure but sometimes also with unmanageably large dimension. In this work we show that adversarial prediction under multivariate losses can be solved much faster than they used to be. We first reduce the problem size *exponentially* by using appropriate sufficient statistics, and then we adapt the new stochastic variance-reduced algorithm of Balamurugan & Bach (2016) to allow any Bregman divergence. We prove that the same linear rate of convergence is retained and we show that for adversarial prediction using KL-divergence we can further achieve a speedup of #example times compared with the Euclidean alternative. We verify the theoretical findings through extensive experiments on two example applications: adversarial prediction and LPboosting.

## 1 Introduction

Many algorithmic advances have been achieved in machine learning by finely leveraging the *separability* in the model. For example, stochastic gradient descent (SGD) algorithms typically exploit the fact that the objective is an expectation of a random function, with each component corresponding to a training example. A "dual" approach partitions the problem into blocks of coordinates and processes them in a stochastic fashion [1]. Recently, by exploiting the finite-sum structure of the model, variance-reduction based stochastic methods have surpassed the well-known sublinear lower bound of SGD. Examples include SVRG [2], SAGA [3], SAG [4], Finito [5], MISO [6], and SDCA [7, 8], just to name a few. Specialized algorithms have also been proposed for accommodating proximal terms [9], and for further acceleration through the condition number [10–13].

However, not all empirical risks are separable in its plain form, and in many cases dualization is necessary for achieving separability. This leads to a composite saddle-point problem with convex-concave (saddle) functions $K$ and $M$:

$$(x^*, y^*) = \arg\min_x \max_y K(x, y) + M(x, y), \text{ where } K(x, y) = \frac{1}{n} \sum_{k=1}^n \psi_k(x, y). \quad (1)$$

Most commonly used supervised losses for linear models can be written as $g^\star(X\mathbf{w})$, where $g^\star$ is the Fenchel dual of a convex function $g$, $X$ is the design matrix, and $\mathbf{w}$ is the model vector. So the regularized risk minimization can be naturally written as $\min_\mathbf{w} \max_{\boldsymbol{\alpha}} \boldsymbol{\alpha}' X\mathbf{w} + \Omega(\mathbf{w}) - g(\boldsymbol{\alpha})$, where $\Omega$ is a regularizer. This fits into our framework (1) with a bilinear function $K$ and a decoupled function $M$. Optimization for this specific form of saddle-point problems has been extensively studied. For example, [14] and [15] performed batch updates on $\mathbf{w}$ and stochastic updates on $\boldsymbol{\alpha}$, while [16] and [17] performed *doubly* stochastic updates on *both* $\mathbf{w}$ and $\boldsymbol{\alpha}$, achieving $O(\frac{1}{\epsilon})$ and $O(\log \frac{1}{\epsilon})$ rates respectively. The latter two also studied the more general form (1). Our interest in this paper is double stochasticity, aiming to maximally harness the power of separability and stochasticity.

Adversarial machines, where the learner competes against an adversary, have re-gained much recent interest in machine learning [18–20]. On one hand they fit naturally into the saddle-point optimization framework (1) but on the other hand they are known to be notoriously challenging to solve. The central message of this work is that certain adversarial machines can be solved significantly faster than they used to be. Key to our development is a new extension of the stochastic variance-reduced algorithm in [17] such that it is compatible with any Bregman divergence, hence opening the possibility to largely reduce the *quadratic* condition number in [17] by better adapting to the underlying geometry using non-Euclidean norms and Bregman divergences.

Improving condition numbers by Bregman divergence has long been studied in (stochastic, proximal) gradient descent [21, 22]. The best known algorithm is arguably stochastic mirror descent [23], which was extended to saddle-points by [16] and to ADMM by [24]. However, they can only achieve the sublinear rate $O(1/\epsilon)$ (for an $\epsilon$-accurate solution). On the other hand, many recent stochastic variance-reduced methods [2–6, 9, 17] that achieve the much faster linear rate $O(\log 1/\epsilon)$ rely inherently on the Euclidean structure, and their extension to Bregman divergence, although conceptually clear, remains challenging in terms of the analysis. For example, the analysis of [17] relied on the resolvent of monotone operators [25] and is hence restricted to the Euclidean norm. In §2 we extend the notion of Bregman divergence to saddle functions and we prove a new Pythagorean theorem that may be of independent interest for analyzing first order algorithms. In §4 we introduce a fundamentally different proof technique (details relegated to Appendix C) that overcomes several challenges arising from a general Bregman divergence (e.g. asymmetry and unbounded gradient on bounded domain), and we recover similar quantitative linear rate of convergence as [17] but with the flexibility of using suitable Bregman divergences to reduce the condition number.

The new stochastic variance-reduced algorithm Breg-SVRG is then applied to the adversarial prediction framework (with multivariate losses such as F-score) [19, 20]. Here we make three novel contributions: (a) We provide a significant reformulation of the adversarial prediction problem that reduces the dimension of the optimization variable from $2^n$ to $n^2$ (where $n$ is the number of samples), hence making it amenable to stochastic variance-reduced optimization (§3). (b) We develop a new efficient algorithm for computing the proximal update with a separable saddle KL-divergence (§5). (c) We verify that Breg-SVRG accelerates its Euclidean alternative by a factor of $n$ in both theory and practice (§6), hence confirming again the uttermost importance of adapting to the underlying problem geometry. To our best knowledge, this is the first time stochastic variance-reduced methods have been shown with great promise in optimizing adversarial machines.

Finally, we mention that we expect our algorithm Breg-SVRG to be useful for solving many other saddle-point problems, and we provide a second example (LPboosting) in experiments (§6).

## 2 Bregman Divergence and Saddle Functions

In this section we set up some notations, recall some background materials, and extend Bregman divergences to saddle functions, a key notion in our later analysis.

**Bregman divergence.** For any convex and differentiable function $\psi$ over some closed convex set $C \subseteq \mathbb{R}^d$, its induced Bregman divergence is defined as:

$$\forall x \in \text{int}(C), x' \in C, \ \ \Delta_\psi(x', x) := \psi(x') - \psi(x) - \langle \nabla \psi(x), x' - x \rangle, \tag{2}$$

where $\nabla \psi$ is the gradient and $\langle \cdot, \cdot \rangle$ is the standard inner product in $\mathbb{R}^d$. Clearly, $\Delta_\psi(x', x) \geq 0$ since $\psi$ is convex. We mention two familiar examples of Bregman divergence.

- Squared Euclidean distance: $\Delta_\psi(x', x) = \frac{1}{2} \|x' - x\|_2^2$, $\psi(x) = \frac{1}{2} \|x\|_2^2$, where $\|\cdot\|_2$ is $\ell_2$ norm.
- (Unnormalized) KL-divergence: $\Delta_\psi(x', x) = \sum_i x'_i \log \frac{x'_i}{x_i} - x'_i + x_i$, $\psi(x) = \sum_i x_i \log x_i$.

**Strong convexity.** Following [26] we call a function $f$ $\psi$-convex if $f - \psi$ is convex, i.e. for all $x, x'$

$$f(x') \geq f(x) + \langle \partial f(x), x' - x \rangle + \Delta_\psi(x', x). \tag{3}$$

**Smoothness.** A function $f$ is $L$-smooth wrt a norm $\|\cdot\|$ if its gradient $\nabla f$ is $L$-Lipschitz continuous, i.e., for all $x$ and $x'$, $\|\nabla f(x') - \nabla f(x)\|_* \leq L \|x' - x\|$, where $\|\cdot\|_*$ is the dual norm of $\|\cdot\|$. The change of a smooth function, in terms of its induced Bregman divergence, can be upper bounded by the change of its input and lower bounded by the change of its slope, cf. Lemma 2 in Appendix A.

**Saddle functions.** Recall that a function $\phi(x, y)$ over $C_z = C_x \times C_y$ is called a saddle function if it is convex in $x$ for any $y \in C_y$, and concave in $y$ for any $x \in C_x$. Given a saddle function $\phi$, we call $(x^*, y^*)$ its saddle point if

$$\forall x \in C_x, \; \forall y \in C_y, \quad \phi(x^*, y) \leq \phi(x^*, y^*) \leq \phi(x, y^*), \tag{4}$$

or equivalently $(x^*, y^*) \in \arg\min_{x \in C_x} \max_{y \in C_y} \phi(x, y)$. Assuming $\phi$ is differentiable, we denote

$$\mathsf{G}_\phi(x, y) := [\partial_x \phi(x, y); -\partial_y \phi(x, y)]. \tag{5}$$

Note the negation sign due to the concavity in $y$. We can quantify the notion of "saddle": A function $f(x, y)$ is called $\phi$-saddle iff $f - \phi$ is a saddle function, or equivalently, $\Delta_f(z', z) \geq \Delta_\phi(z', z)$ (see below). Note that any saddle function $\phi$ is 0-saddle and $\phi$-saddle.

**Bregman divergence for saddle functions.** We now define the Bregman divergence induced by a saddle function $\phi$: for $z = (x, y)$ and $z' = (x', y')$ in $C_z$,

$$\Delta_\phi(z', z) := \Delta_{\phi_y}(x', x) + \Delta_{-\phi_x}(y', y) = \phi(x', y) - \phi(x, y') - \langle \mathsf{G}_\phi(z), z' - z \rangle, \tag{6}$$

where $\phi_y(x) = \phi(x, y)$ is a convex function of $x$ for any fixed $y$, and similarly $\phi_x(y) = \phi(x, y)$ is a concave (hence the negation) function of $y$ for any fixed $x$. The similarity between (6) and the usual Bregman divergence $\Delta_\psi$ in (2) is apparent. However, $\phi$ is never evaluated at $z'$ but $z$ (for $\mathsf{G}$) and the cross pairs $(x', y)$ and $(x, y')$. Key to our subsequent analysis is the following lemma that extends a result of [27] to saddle functions (proof in Appendix A).

**Lemma 1.** *Let $f$ and $g$ be $\phi$-saddle and $\varphi$-saddle respectively, with one of them being differentiable. Then, for any $z = (x, y)$ and any saddle point (if exists) $z^* := (x^*, y^*) \in \arg\min_x \max_y \{f(z) + g(z)\}$, we have $f(x, y^*) + g(x, y^*) \geq f(x^*, y) + g(x^*, y) + \Delta_{\phi+\varphi}(z, z^*)$.*

**Geometry of norms.** In the sequel, we will design two convex functions $\psi_x(x)$ and $\psi_y(y)$ such that their induced Bregman divergences are "distance enforcing" (a.k.a. 1-strongly convex), that is, w.r.t. two norms $\|\cdot\|_x$ and $\|\cdot\|_y$ that we also design, the following inequality holds:

$$\Delta_x(x, x') := \Delta_{\psi_x}(x, x') \geq \tfrac{1}{2} \|x - x'\|_x^2, \; \Delta_y(y, y') := \Delta_{\psi_y}(y, y') \geq \tfrac{1}{2} \|y - y'\|_y^2. \tag{7}$$

Further, for $z = (x, y)$, we define

$$\Delta_z(z, z') := \Delta_{\psi_x - \psi_y}(z, z') \geq \tfrac{1}{2} \|z - z'\|_z^2, \quad \text{where} \quad \|z\|_z^2 := \|x\|_x^2 + \|y\|_y^2 \tag{8}$$

When it is clear from the context, we simply omit the subscripts and write $\Delta$, $\|\cdot\|$, and $\|\cdot\|_*$.

## 3 Adversarial Prediction under Multivariate Loss

A number of saddle-point based machine learning problems have been listed in [17]. Here we give another example (adversarial prediction under multivariate loss) that is naturally formulated as a saddle-point problem but also requires a careful adaptation to the underlying geometry—a challenge that was not addressed in [17] since their algorithm inherently relies on the Euclidean norm. We remark that adaptation to the underlying geometry has been studied in the (stochastic) mirror descent framework [23], with significant improvements on condition numbers or gradient norm bounds. Surprisingly, no analogous efforts have been attempted in the stochastic variance reduction framework—a gap we intend to fill in this work.

The adversarial prediction framework [19, 20, 28], arising naturally as a saddle-point problem, is a convex alternative to the generative adversarial net [18]. Given a training sample $X = [\mathbf{x}_1, \dots, \mathbf{x}_n]$ and $\tilde{\mathbf{y}} = [\tilde{y}_1, \dots, \tilde{y}_n] \in \{0, 1\}^n$, adversarial prediction optimizes the following saddle function that is an expectation of some multivariate loss $\ell(\mathbf{y}, \mathbf{z})$ (e.g. F-score) over the labels $\mathbf{y}, \mathbf{z} \in \{0, 1\}^n$ of all data points:

$$\min_{p \in \Delta^{2^n}} \left[ \max_{q \in \Delta^{2^n}} \; \mathbb{E}_{\mathbf{y} \sim p, \mathbf{z} \sim q} \; \ell(\mathbf{y}, \mathbf{z}), \; \text{s.t.} \; \mathbb{E}_{\mathbf{z} \sim q} \left( \tfrac{1}{n} X \mathbf{z} \right) = \tfrac{1}{n} X \tilde{\mathbf{y}} \right] \tag{9}$$

Here the proponent tries to find a distribution $p(\cdot)$ over the labeling on the entire training set in order to minimize the loss ($\Delta^{2^n}$ is the $2^n$ dimensional probability simplex). An opponent in contrast tries to maximize the expected loss by finding another distribution $q(\cdot)$, but his strategy is subject to the constraint that the feature expectation matches that of the empirical distribution. Introducing a

Lagrangian variable $\boldsymbol{\theta}$ to remove the feature expectation constraint and specializing the problem to F-score where $\ell(\mathbf{y}, \mathbf{z}) = \frac{2\mathbf{y}'\mathbf{z}}{\mathbf{1}'\mathbf{y}+\mathbf{1}'\mathbf{z}}$ and $\ell(\mathbf{0}, \mathbf{0}) := 1$, the partial dual problem can be written as

$$\max_{\boldsymbol{\theta}} \ -\frac{\lambda}{2}\|\boldsymbol{\theta}\|_2^2 + \frac{1}{n}\boldsymbol{\theta}'X\tilde{\mathbf{y}} + \min_{p\in\Delta^{2^n}}\max_{q\in\Delta^{2^n}} \mathbb{E}_{\mathbf{y}\sim p, \mathbf{z}\sim q}\left[\frac{2\mathbf{y}'\mathbf{z}}{\mathbf{1}'\mathbf{y}+\mathbf{1}'\mathbf{z}} - \frac{1}{n}\boldsymbol{\theta}'X\mathbf{y}\right], \tag{10}$$

where we use $\mathbf{y}'\mathbf{z}$ to denote the standard inner product and we followed [19] to add an $\ell_2^2$ regularizer on $\boldsymbol{\theta}$ penalizing the dual variables on the constraints over the training data. It appears that solving (10) can be quite challenging, because the variables $p$ and $q$ in the inner minimax problem have $2^n$ entries! A constraint sampling algorithm was adopted in [19] to address this challenge, although no formal guarantee was established. Note that we can maximize the outer unconstrained variable $\boldsymbol{\theta}$ (with dimension the same as the number of features) relatively easily using for instance gradient ascent, provided that we can solve the inner minimax problem quickly—a significant challenge to which we turn our attention below.

Surprisingly, we show here that the inner minimax problem in (10) can be significantly simplified. The key observation is that the expectation in the objective depends only on a few sufficient statistics of $p$ and $q$. Indeed, by interpreting $p$ and $q$ as probability distributions over $\{0,1\}^n$ we have:

$$\mathbb{E}\frac{2\mathbf{y}'\mathbf{z}}{\mathbf{1}'\mathbf{y} + \mathbf{1}'\mathbf{z}} = p(\{\mathbf{0}\})q(\{\mathbf{0}\}) + \sum_{i=1}^{n}\sum_{j=1}^{n}\mathbb{E}\left(\frac{2\mathbf{y}'\mathbf{z}}{\mathbf{1}'\mathbf{y}+\mathbf{1}'\mathbf{z}}[\![\mathbf{1}'\mathbf{y}=i]\!][\![\mathbf{1}'\mathbf{z}=j]\!]\right) \tag{11}$$

$$= p(\{\mathbf{0}\})q(\{\mathbf{0}\}) + \sum_{i=1}^{n}\sum_{j=1}^{n}\frac{2ij}{i+j}\cdot\underbrace{\frac{1}{i}\mathbb{E}\left(\mathbf{y}[\![\mathbf{1}'\mathbf{y}=i]\!]\right)'}_{\boldsymbol{\alpha}_i}\cdot\underbrace{\frac{1}{j}\mathbb{E}\left(\mathbf{z}[\![\mathbf{1}'\mathbf{z}=j]\!]\right)}_{\boldsymbol{\beta}_j}, \tag{12}$$

where $[\![\cdot]\!] = 1$ if $\cdot$ is true, and 0 otherwise. Crucially, the variables $\boldsymbol{\alpha}_i$ and $\boldsymbol{\beta}_j$ are sufficient for re-expressing (10), since

$$\mathbf{1}'\boldsymbol{\alpha}_i = \frac{1}{i}\mathbb{E}\left(\mathbf{1}'\mathbf{y}[\![\mathbf{1}'\mathbf{y}=i]\!]\right) = \mathbb{E}[\![\mathbf{1}'\mathbf{y}=i]\!] = p(\{\mathbf{1}'\mathbf{y}=i\}), \tag{13}$$

$$\sum_i i\boldsymbol{\alpha}_i = \sum_i \mathbb{E}\left(\mathbf{y}[\![\mathbf{1}'\mathbf{y}=i]\!]\right) = \mathbb{E}\mathbf{y}, \tag{14}$$

and similar equalities also hold for $\boldsymbol{\beta}_j$. In details, the inner minimax problem of (10) simplifies to:

$$\min_{\boldsymbol{\alpha}\in S}\max_{\boldsymbol{\beta}\in S} \frac{1}{n^2}\sum_{i=1}^{n}\sum_{j=1}^{n}\Big[\underbrace{\frac{2ijn^2}{i+j}\boldsymbol{\alpha}_i'\boldsymbol{\beta}_j + n^2\boldsymbol{\alpha}_i'\mathbf{1}\mathbf{1}'\boldsymbol{\beta}_j}_{f_{ij}(\boldsymbol{\alpha}_i,\boldsymbol{\beta}_j)} - n\mathbf{1}'\boldsymbol{\alpha}_i - n\mathbf{1}'\boldsymbol{\beta}_j - \boldsymbol{\theta}'Xi\boldsymbol{\alpha}_i\Big] + \Omega(\boldsymbol{\alpha}) - \Omega(\boldsymbol{\beta}), \tag{15}$$

$$\text{where } S = \{\boldsymbol{\alpha} \geq \mathbf{0} : \mathbf{1}'\boldsymbol{\alpha} \leq 1, \forall i, \|i\boldsymbol{\alpha}_i\|_\infty \leq \|\boldsymbol{\alpha}_i\|_1\}, \quad \Omega(\boldsymbol{\alpha}) = \mu\sum_{i,j}\alpha_{ij}\log(\alpha_{ij}). \tag{16}$$

Importantly, $\boldsymbol{\alpha} = [\boldsymbol{\alpha}_1; \ldots, \boldsymbol{\alpha}_n]$ (resp. $\boldsymbol{\beta}$) has $n^2$ entries, which is significantly smaller than the $2^n$ entries of $p$ (resp. $q$) in (10). For later purpose we have also incorporated an entropy regularizer for $\boldsymbol{\alpha}$ and $\boldsymbol{\beta}$ respectively in (15).

To justify the constraint set $S$, note from (12) and (13) that for any distribution $p$ of $\mathbf{y}$:

$$\text{since } \boldsymbol{\alpha} \geq 0 \text{ and } \mathbf{y} \in \{0,1\}^n, \ \|i\boldsymbol{\alpha}_i\|_\infty \leq \mathbb{E}\|\mathbf{y}[\![\mathbf{1}'\mathbf{y}=i]\!]\|_\infty \leq \mathbb{E}[\![\mathbf{1}'\mathbf{y}=i]\!] = \|\boldsymbol{\alpha}_i\|_1. \tag{17}$$

Conversely, for any $\boldsymbol{\alpha} \in S$, we can construct a distribution $p$ such that $i\alpha_{ij} = \mathbb{E}\left(y_j[\![\mathbf{1}'\mathbf{y}=i]\!]\right) = p(\{\mathbf{1}'\mathbf{y}=i, y_j=1\})$ in the following algorithmic way: Fix $i$ and for each $j$ define $Y_j = \{\mathbf{y} \in \{0,1\}^n : \mathbf{1}'\mathbf{y}=i, y_j=1\}$. Let $U = \{1, \ldots, n\}$. Find an index $j$ in $U$ that minimizes $\alpha_{ij}$ and set $p(\{\mathbf{y}\}) = i\alpha_{ij}/|Y_j|$ for each $\mathbf{y} \in Y_j$. Perform the following updates:

$$U \leftarrow U \setminus \{j\}, \ \forall k \neq j, Y_k \leftarrow Y_k \setminus Y_j, \ \alpha_{ik} \leftarrow \alpha_{ik} - \alpha_{ij}|Y_k \cap Y_j|/|Y_j| \tag{18}$$

Continue this procedure until $U$ is empty. Due to the way we choose $j$, $\boldsymbol{\alpha}$ remains nonnegative and by construction $\alpha_{ij} = p(\{\mathbf{1}'\mathbf{y}=i, y_j=1\})$ once we remove $j$ from $U$.

The objective function in (15) fits naturally into the framework of (1), with $\Omega(\boldsymbol{\alpha}) - \Omega(\boldsymbol{\beta})$ and constraints corresponding to $M$, and the rest terms to $K$. The entropy function $\Omega$ is convex wrt the KL-divergence, which is in turn distance enforcing wrt the $\ell_1$ norm over the probability simplex [23]. In the next section we propose the SVRG algorithm with Bregman divergence (Breg-SVRG) that (a) provably optimizes strongly convex saddle function with a linear convergence rate, and (b) adapts to the underlying geometry by choosing an appropriate Bregman divergence. Then, in §5 we apply Breg-SVRG to (15) and achieve a factor of $n$ speedup over a straightforward instantiation of [17].

## 4 Breg-SVRG for Saddle-Point

In this section we propose an efficient algorithm for solving the general saddle-point problem in (1) and prove its linear rate of convergence. Our main assumption is:

**Assumption 1.** *There exist two norms $\|\cdot\|_{\mathsf{x}}$ and $\|\cdot\|_{\mathsf{y}}$ such that each $\psi_k$ is a saddle function and $L$-smooth; $M$ is $(\psi_{\mathsf{x}} - \psi_{\mathsf{y}})$-saddle; and $\psi_{\mathsf{x}}$ and $\psi_{\mathsf{y}}$ are distance enforcing (cf. (7))*.

Note that w.l.o.g. we have scaled the norms so that the usual strong convexity parameter of $M$ is 1.

---

**Algorithm 1:** Breg-SVRG for Saddle-Point

**1** Initialize $z_0$ randomly. Set $\tilde{z} = z_0$.
**2** **for** $s = 1, 2, \ldots$ **do** ▷ `epoch index`
**3**     $\tilde{\mu} \leftarrow \tilde{\mu}^s := \nabla K(\tilde{z})$, $z_0 \leftarrow z_0^s := z_m$
**4**     **for** $t = 1, \ldots, m$ **do** ▷ `iter index`
**5**        Randomly pick $\xi \in \{1, \ldots, n\}$.
**6**        Compute $v_t$ using (20).
**7**        Update $z_t$ using (21).
**8**     $\tilde{z} \leftarrow \tilde{z}^s := \sum_{t=1}^{m}(1+\eta)^t z_t \Big/ \sum_{t=1}^{m}(1+\eta)^t$.

---

Recall we defined $\|z\|_{\mathsf{z}}$ and $\Delta_{\mathsf{z}}$ in (8). For saddle-point optimization, it is common to define a signed gradient $\mathsf{G}(z) := [\partial_x K(z); -\partial_y K(z)]$ (since $K$ is concave in $y$). Recall $J = K + M$, and $(x^*, y^*)$ is a saddle-point of $J$. Using Assumption 1, we measure the gap of an iterate $z_t = (x_t, y_t)$ as follows:

$$\epsilon_t = \epsilon(z_t) = J(x_t, y^*) - J(x^*, y_t) \geq \Delta(z_t, z^*) \geq \tfrac{1}{2}\|z_t - z^*\|^2 \geq 0. \tag{19}$$

Inspired by [2, 9, 17], we propose in Algorithm 1 a new stochastic variance-reduced algorithm for solving the saddle-point problem (1) using Bregman divergences. The algorithm proceeds in epochs. In each epoch, we first compute the following stochastic estimate of the signed gradient $G(z_t)$ by drawing a random component from $K$:

$$v_t = \begin{pmatrix} v_x(z_t) \\ -v_y(z_t) \end{pmatrix} \quad \text{where} \quad \begin{cases} v_x(z_t) := \partial_x \psi_\xi(z_t) - \partial_x \psi_\xi(\tilde{z}) + \partial_x K(\tilde{z}) \\ v_y(z_t) := \partial_y \psi_\xi(z_t) - \partial_y \psi_\xi(\tilde{z}) + \partial_y K(\tilde{z}) \end{cases}. \tag{20}$$

Here $\tilde{z}$ is the pivot chosen after completing the previous epoch. We make two important observations: (1) By construction the stochastic gradient $v_t$ is unbiased: $\mathbb{E}_\xi[v_t] = \mathsf{G}(z_t)$; (2) The expensive gradient evaluation $\partial K(\tilde{z})$ need only be computed once in each epoch since $\tilde{z}$ is held unchanged. If $\tilde{z} \to z^*$, then the variance of $v_t$ would be largely reduced hence faster convergence may be possible.

Next, Algorithm 1 performs the following *joint* proximal update:

$$(x_{t+1}, y_{t+1}) = \arg\min_x \max_y \eta \langle v_x(z_t), x \rangle + \eta \langle v_y(z_t), y \rangle + \eta M(x, y) + \Delta(x, x_t) - \Delta(y, y_t), \tag{21}$$

where we have the flexibility in choosing a suitable Bregman divergence to better adapt to the underlying geometry. When $\Delta(x, x_t) = \frac{1}{2}\|x - x_t\|_2^2$, we recover the special case in [17]. However, to handle the asymmetry in a general Bregman divergence (which does not appear for the Euclidean distance), we have to choose the pivot $\tilde{z}$ in a significantly different way than [2, 9, 17].

We are now ready to present our main convergence guarantee for Breg-SVRG in Algorithm 1.

**Theorem 1.** *Let Assumption 1 hold, and choose a sufficiently small $\eta > 0$ such that $m := \left\lceil \log\left(\frac{1-\eta L}{18\eta L^2} - \eta - 1\right) \Big/ \log(1+\eta) \right\rceil \geq 1$. Then Breg-SVRG enjoys linear convergence in expectation:*

$$\mathbb{E}\epsilon(\tilde{z}^s) \leq (1+\eta)^{-ms}[\Delta(z^*, z_0) + c(Z+1)\epsilon(z_0)], \text{ where } Z = \sum_{t=0}^{m-1}(1+\eta)^t, \ c = \frac{18\eta^2 L^2}{1-\eta L}. \tag{22}$$

For example, we may set $\eta = \frac{1}{45L^2}$, which leads to $c = O(1/L^2)$, $m = \Theta\left(L^2\right)$, $(1+\eta)^m \geq \frac{64}{45}$, and $Z = O(L^2)$. Therefore, between epochs, the gap $\epsilon(\tilde{z}^s)$ decays (in expectation) by a factor of $\frac{45}{64}$, and each epoch needs to conduct the proximal update (21) for $m = \Theta(L^2)$ number of times. (We remind that w.l.o.g. we have scaled the norms so that the usual strong convexity parameter is 1.) In total, to reduce the gap below some threshold $\epsilon$, Breg-SVRG needs to call the proximal update (21) $O(L^2 \log \frac{1}{\epsilon})$ number of times, plus a similar number of *component* gradient evaluations.

**Discussions.** As mentioned, Algorithm 1 and Theorem 1 extend those in [17] which in turn extend [2, 9] to saddle-point problems. However, [2, 9, 17] all heavily exploit the Euclidean structure (in particular symmetry) hence their proofs cannot be applied to an asymmetric Bregman divergence. Our innovations here include: (a) A new Pythagorean theorem for the newly introduced saddle Bregman divergence (Lemma 1). (b) A moderate extension of the variance reduction lemma in [9] to accommodate any norm (Appendix B). (c) A different pivot $\tilde{z}$ is adopted in each epoch to handle

asymmetry. (d) A new analysis technique through introducing a crucial auxiliary variable that enables us to bound the function gap directly. See our proof in Appendix C for more details. Compared with classical mirror descent algorithms [16, 23] that can also solve saddle-point problems with Bregman divergences, our analysis is fundamentally different and we achieve the significantly stronger rate $O(\log(1/\epsilon))$ than the sublinear $O(1/\epsilon)$ rate of [16], at the expense of a squared instead of linear dependence on $L$. Similar tradeoff also appeared in [17]. We will return to this issue in Section 5.

**Variants and acceleration.** Our analysis also supports to use different $\xi$ in $v_x$ and $v_y$. The standard acceleration methods such as universal catalyst [10] and non-uniform sampling can be applied directly (see Appendix E where $L$, the largest smoothness constant over all pieces, is replaced by their mean).

## 5 Application of Breg-SVRG to Adversarial Prediction

The quadratic dependence on $L$, the smoothness parameter, in Theorem 1 reinforces the need to choose suitable Bregman divergences. In this section we illustrate how this can be achieved for the adversarial prediction problem in Section 3. As pointed out in [17], the factorization of $K$ is important, and we consider three schemes: (a) $\psi_k = f_{ij}$; (b) $\psi_k = \frac{1}{n}\sum_{j=1}^{n} f_{k,j}$; and (c) $\psi_k = \frac{1}{n}\sum_{i=1}^{n} f_{i,k}$. W.l.o.g. let us fix the $\mu$ in (16) to 1.

**Comparison of smoothness constant.** Both $\boldsymbol{\alpha}$ and $\boldsymbol{\beta}$ are $n^2$-dimensional, and the bilinear function $f_{ij}$ can be written as $\boldsymbol{\alpha}' A_{ij} \boldsymbol{\beta}$, where $A_{ij} \in \mathbb{R}^{n^2 \times n^2}$ is an $n$-by-$n$ block matrix, with the $(i,j)$-th block being $n^2(\frac{2ij}{i+j} I + \mathbf{1}\mathbf{1}')$ and all other blocks being $\mathbf{0}$. The linear terms in (15) can be absorbed into the regularizer $\Omega$ without affecting the smoothness parameter.

For scheme (a), the smoothness constant $L_2$ under $\ell_2$ norm depends on the spectral norm of $A_{ij}$: $L_2 = \max_{i,j} n^2(n + \frac{2ij}{i+j}) = \Theta(n^3)$. In contrast the smoothness constant $L_1$ under $\ell_1$ norm depends on the absolute value of the entries in $A_{ij}$: $L_1 = \max_{i,j} n^2(1 + \frac{2ij}{i+j}) = \Theta(n^3)$; no saving is achieved.

For scheme (b), the bilinear function $\psi_k$ corresponds to $\frac{1}{n}\boldsymbol{\alpha}'\sum_{j=1}^{n} A_{kj}\boldsymbol{\beta}$. Then $L_1 = O(n^2)$ while

$$L_2^2 = \frac{1}{n^2} \max_k \max_{\mathbf{v}:\|\mathbf{v}\|_2=1} \sum_{j=1}^{n} \|A_{kj}\mathbf{v}\|_2^2 \geq n^2 \max_{\|\mathbf{v}\|_2=1} \sum_{j=1}^{n} \|\mathbf{1}\mathbf{1}'\mathbf{v}\|^2 = n^5. \qquad (23)$$

Therefore, $L_1^2$ saves a factor of $n$ compared with $L_2^2$.

**Comparison of smoothness constant for the overall problem.** By strong duality, we may push the maximization over $\boldsymbol{\theta}$ to the innermost level of (10), arriving at an overall problem in $\boldsymbol{\alpha}$ and $\boldsymbol{\beta}$ only:

$$\min_{\{\boldsymbol{\alpha}_i\}\in S} \max_{\{\boldsymbol{\beta}_j\}\in S} \frac{1}{n^2} \sum_{i=1}^{n} \sum_{j=1}^{n} \left[ f_{ij}(\boldsymbol{\alpha}_i, \boldsymbol{\beta}_j) - \frac{i}{\lambda n}\mathbf{c}'X\boldsymbol{\alpha}_i + \frac{ij}{2\lambda}\boldsymbol{\alpha}_i'X'X\boldsymbol{\alpha}_j + \frac{1}{2\lambda n^2}\|\mathbf{c}\|_2^2 \right]. \qquad (24)$$

where $\mathbf{c} = X\tilde{\mathbf{y}}$. The quadratic term w.r.t. $\boldsymbol{\alpha}$ can be written as $\boldsymbol{\alpha}' B_{ij}\boldsymbol{\alpha}$, where $B_{ij} \in \mathbb{R}^{n^2 \times n^2}$ is an $n$-by-$n$ block matrix, with its $(i,j)$-th block being $\frac{ij}{2\lambda}X'X$ and all other blocks being $\mathbf{0}$. And we assume each $\|\mathbf{x}_i\|_2 \leq 1$. The smoothness constant can be bounded separately from $A_{ij}$ and $B_{ij}$; see (128) in Appendix F.

For scheme (a), the smoothness constant square $L_2^2$ under $\ell_2$ norm is upper bounded by the sum of spectral norm square of $A_{ij}$ and $B_{ij}$. So $L_2^2 \geq \max_{i,j}\left(\frac{ij}{2\lambda}n\right)^2 = \Omega(n^6)$, i.e. $L_2 = \Theta(n^3)$. In contrast the smoothness constant square $L_1^2$ under $\ell_1$ norm is at most the sum of square of maximum absolute value of the entries in $A_{ij}$ and $B_{ij}$. Hence $L_1^2 \leq \max_{i,j}\left(n^2(1 + \frac{2ij}{i+j})\right)^2 + \max_{i,j}\left(\frac{ij}{2\lambda}\right)^2 = \Theta(n^6)$, i.e. $L_1 = \Theta(n^3)$. So no saving is achieved here.

For scheme (b), $\psi_k$ corresponds to $\frac{1}{n}(\boldsymbol{\alpha}'\sum_{j=1}^{n} A_{kj}\boldsymbol{\beta} + \boldsymbol{\alpha}'\sum_{j=1}^{n} B_{kj}\boldsymbol{\alpha})$. Then

$$L_1^2 \leq \frac{1}{n^2} \max_k \left[ \max_{\mathbf{v}:\|\mathbf{v}\|_1=1} \|\sum_{j=1}^{n} A_{kj}\mathbf{v}\|_\infty^2 + \max_{\mathbf{v}:\|\mathbf{v}\|_1=1} \|\sum_{j=1}^{n} B_{kj}\mathbf{v}\|_\infty^2 \right] \quad \text{(by (128))} \qquad (25)$$

$$\leq \frac{1}{n^2} \max_k \max_j \left[ \left(n^2(1 + \frac{2kj}{k+j})\right)^2 + \left(\frac{kj}{2}\right)^2 \right] = n^4, \qquad (26)$$

and by setting $\boldsymbol{\beta}$ to $\mathbf{0}$ in (126), we get $L_2^2 \geq n^5$ similar to (23). Therefore, $L_1^2$ saves a factor of $n$ compared with $L_2^2$. Similar results apply to scheme (c) too. We also tried non-uniform sampling, but

it does not change the order in $n$. It can also be shown that if our scheme randomly samples $n$ entries from $\{A_{ij}, B_{ij}\}$, the above $L_1$ and $L_2$ cannot be improved by further engineering the factorization.

**Computational complexity.** We finally seek efficient algorithms for the proximal update (21) used by Breg-SVRG. When $M(\boldsymbol{\alpha}, \boldsymbol{\beta}) = \Omega(\boldsymbol{\alpha}) - \Omega(\boldsymbol{\beta})$ as in (16), we can solve $\boldsymbol{\alpha}$ and $\boldsymbol{\beta}$ separately as:

$$\min_{\boldsymbol{\alpha}} \ \sum_{ik} \alpha_{ik} \log(\alpha_{ik}/b_{ik}) - c_{ik}, \quad \text{s.t.} \quad \mathbf{1}'\boldsymbol{\alpha} \leq 1, \ \forall i \ \forall k, \ 0 \leq i\alpha_{ik} \leq \mathbf{1}'\boldsymbol{\alpha}_i. \tag{27}$$

where $b_{ik}$ and $c_{ik}$ are constants. In Appendix D we designe an efficient "closed form" algorithm which finds an $\epsilon$ accurate solution in $O(n^2 \log^2 \frac{1}{\epsilon})$ time, which is also on par with that for computing the stochastic gradient in schemes (b) and (c). Although scheme (a) reduces the cost of gradient computation to $O(n)$, its corresponding smoothness parameter $L_1^2$ is increased by $n^2$ times, hence not worthwhile. We did manage to design an $\tilde{O}(n)$ algorithm for the proximal update in scheme (a), but empirically the overall convergence is rather slow.

If we use the Euclidean squared distance as the Bregman divergence, then a term $\|\boldsymbol{\alpha} - \boldsymbol{\alpha}_t\|_2^2$ needs to be added to the objective (27). No efficient "closed form" solution is available, and so in experiments we simply absorbed $M$ into $K$, and then the proximal update becomes the Euclidean projection onto $S$, which does admit a competitive $O(n^2 \log^2(1/\epsilon))$ time solution.

# 6   Experimental Results

Our major goal here is to show that empirically Entropy-SVRG (Breg-SVRG with KL divergence) is significantly more efficient than Euclidean-SVRG (Breg-SVRG with squared Euclidean distance) on some learning problems, especially those with an entropic regularizer and a simplex constraint.

## 6.1   Entropy regularized LPBoost

We applied Breg-SVRG to an extension of LP Boosting using entropy regularization [29]. In a binary classification setting, the base hypotheses over the training set can be compactly represented as $U = (y_1\mathbf{x}_1, \ldots, y_n\mathbf{x}_n)'$. Then the model considers a minimax game between a distribution $\mathbf{d} \in \Delta^n$ over training examples and a distribution $\mathbf{w} \in \Delta^m$ over the hypotheses:

$$\min_{\mathbf{d} \in \Delta^n, d_i \leq \nu} \ \max_{\mathbf{w} \in \Delta^m} \mathbf{d}'U\mathbf{w} + \lambda\Omega(\mathbf{d}) - \gamma\Omega(\mathbf{w}). \tag{28}$$

Here $\mathbf{w}$ tries to combine the hypotheses to maximize the edge (prediction confidence) $y_i\mathbf{x}_i'\mathbf{w}$, while the adversary $\mathbf{d}$ tries to place more weights (bounded by $\nu$) on "hard" examples to reduce the edge.

**Settings.** We experimented on the adult dataset from the UCI repository, which we partitioned into $n = 32,561$ training examples and 16,281 test examples, with $m = 123$ features. We set $\lambda = \gamma = 0.01$ and $\nu = 0.1$ due to its best prediction accuracy. We tried a range of values of the step size $\eta$, and the best we found was $10^{-3}$ for Entropy-SVRG and $10^{-6}$ for Euclidean-SVRG (larger step size for Euclidean-SVRG fluctuated even worse). For both methods, $m = 32561/50$ gave good results.

The stochastic gradient in $\mathbf{d}$ was computed by $U_{:j}w_j$, where $U_{:j}$ is the $j$-th column and $j$ is randomly sampled. The stochastic gradient in $\mathbf{w}$ is $d_iU_{i:}'$. We tried with $U_{ij}w_j$ and $U_{ij}d_i$ (scheme (a) in §5), but they performed worse. We also tried with the universal catalyst in the same form as [17], which can be directly extended to Entropy-SVRG. Similarly we used the non-uniform sampling based on the $\ell_2$ norm of the rows and columns of $U$. It turned out that the Euclidean-SVRG can benefit slightly from it, while Entropy-SVRG does not. So we only show the "accelerated" results for the former.

To make the computational cost comparable across machines, we introduced a counter called effective number of passes: #pass. Assume the proximal operator has been called #po number of times, then

$$\#\text{pass} \ := \ \text{number of epochs so far} \ + \frac{n+m}{nm} \cdot \#\text{po}. \tag{29}$$

We also compared with a "convex" approach. Given $\mathbf{d}$, the optimal $\mathbf{w}$ in (28) obviously admits a closed-form solution. General saddle-point problems certainly do not enjoy such a convenience. However, we hope to take advantage of this opportunity to study the following question: suppose we solve (28) as a convex optimization in $\mathbf{d}$ and the stochastic gradient were computed from the optimal

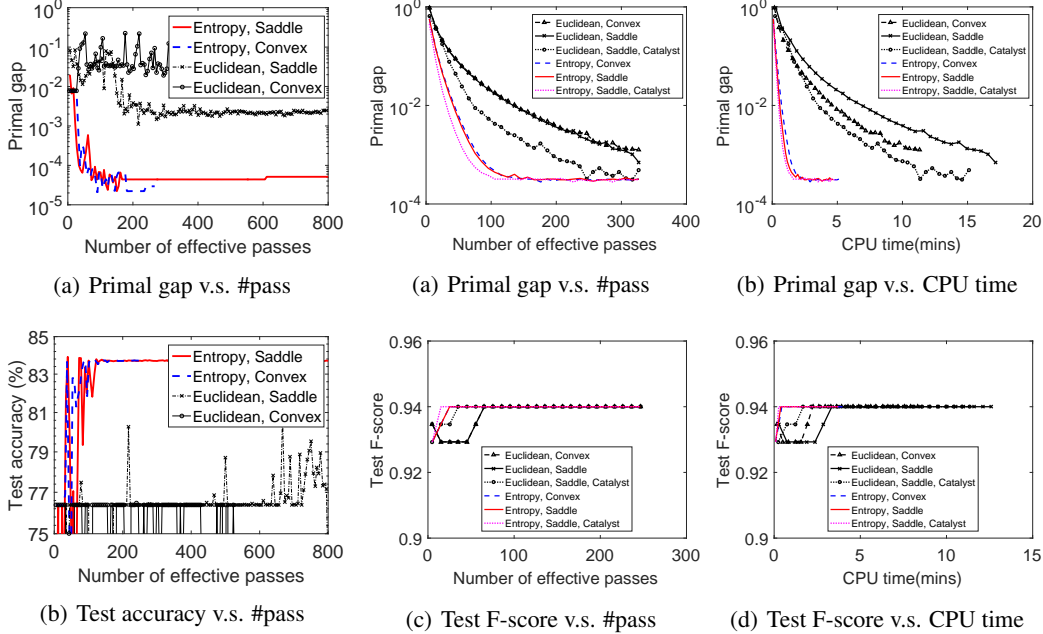

(a) Primal gap v.s. #pass     (a) Primal gap v.s. #pass     (b) Primal gap v.s. CPU time

(b) Test accuracy v.s. #pass     (c) Test F-score v.s. #pass     (d) Test F-score v.s. CPU time

Figure 1: Entropy Regularized LPBoost on adult

Figure 2: Adversarial Prediction on the synthetic dataset.

$\mathbf{w}$, would it be faster than the saddle SVRG? Since solving $\mathbf{w}$ requires visiting the entire $U$, strictly speaking the term $\frac{n+m}{nm}$·#po in the definition of #pass in (29) should be replaced by #po. However, we stuck with (29) because our interest is whether a more accurate stochastic gradient in $\mathbf{d}$ (based on the optimal $\mathbf{w}$) can outperform doubly stochastic (saddle) optimization. We emphasize that this comparison is only for conceptual understanding, because generally optimizing the inner variable requires costly iterative methods.

**Results.** Figure 1(a) demonstrated how fast the *primal gap* (with $\mathbf{w}$ optimized out for each $\mathbf{d}$) is reduced as a function of the number of effective passes. Methods based on entropic prox are clearly much more efficient than Euclidean prox. This corroborates our theory that for problems like (28), Entropy-SVRG is more suitable for the underlying geometry (entropic regularizer with simplex constraints).

We also observed that using entropic prox, our doubly stochastic method is as efficient as the "convex" method, meaning that although at each iteration the $\mathbf{w}$ in saddle SVRG is not the optimal for the current $\mathbf{d}$, it still allows the overall algorithm to perform as fast as if it were. This suggests that for general saddle-point problems where no closed-form inner solution is available, our method will still be efficient and competitive. Note this "convex" method is similar to the optimizer used by [29].

Finally, we investigated the increase of test accuracy as more passes over the data are performed. Figure 1(b) shows, once more, that the entropic prox does allow the accuracy to be improved much faster than Euclidean prox. Again, the convex and saddle methods perform similarly.

As a final note, the Euclidean/entropic proximal operator for both $\mathbf{d}$ and $\mathbf{w}$ can be solved in either closed form, or by a 1-D line search based on partial Lagrangian. So their computational cost differ in the same order of magnitude as multiplication v.s. exponentiation, which is much smaller than the difference of #pass shown in Figure 1.

## 6.2 Adversarial prediction with F-score

**Datasets.** Here we considered two datasets. The first is a synthetic dataset where the positive examples are drawn from a 200 dimensional normal distribution with mean $0.1 \cdot \mathbf{1}$ and covariance $0.5 \cdot I$, and negative examples are drawn from $\mathcal{N}(-0.1 \cdot \mathbf{1}, \ 0.5 \cdot I)$. The training set has $n = 100$ samples, half are positive and half are negative. The test set has 200 samples with the same class ratio. Notice that $n = 100$ means we are optimizing over two 100-by-100 matrices constrained to a challenging set $S$. So the optimization problem is indeed not trivial.

The second dataset, ionosphere, has 211 training examples (122 pos and 89 neg). 89 examples were used for testing (52 pos and 37 neg). Each example has 34 features.

**Methods.** To apply saddle SVRG, we used strong duality to push the optimization over $\theta$ to the inner-most level of (10), and then eliminated $\theta$ because it is a simple quadratic. So we ended up with the convex-concave optimization as shown in (24), where the $K$ part of (15) is augmented with a quadratic term in $\alpha$. The formulae for computing the stochastic gradient using scheme (b) are detailed in Appendix G. We fixed $\mu = 1$, $\lambda = 0.01$ for the ionosphere dataset, and $\mu = 1$, $\lambda = 0.1$ for the synthetic dataset.

We also tried the universal catalyst along with non-uniform sampling where each $i$ was sampled with a probability proportional to $\sum_{k=1}^{n} \|A_{ik}\|_F^2$, and similarly for $j$. Here $\|\cdot\|_F$ is the Frobenious norm.

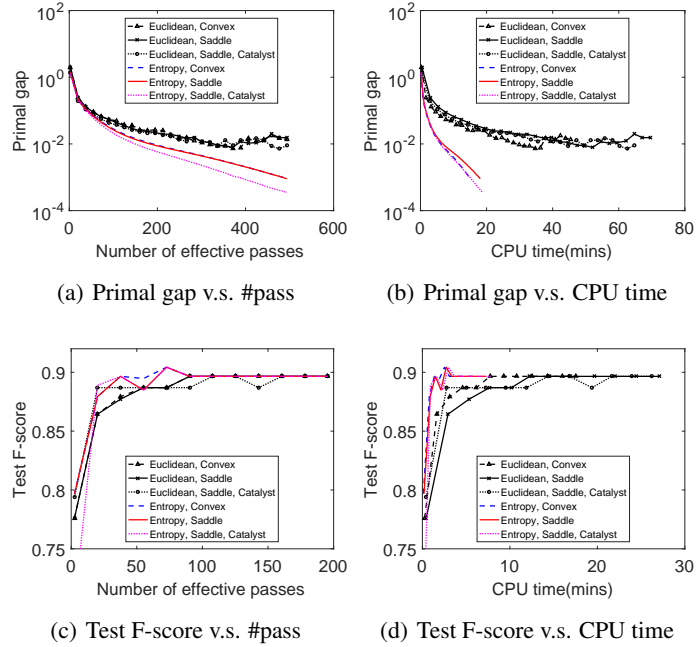

(a) Primal gap v.s. #pass     (b) Primal gap v.s. CPU time

(c) Test F-score v.s. #pass     (d) Test F-score v.s. CPU time

Figure 3: Adversarial Prediction on the ionosphere dataset.

**Parameter Tuning.** Since each entry in the $n \times n$ matrix $\alpha$ is relatively small when $n$ is large, we needed a relatively small step size. When $n = 100$, we used $10^{-2}$ for Entropy-SVRG and $10^{-6}$ for Euclidean-SVRG (a larger step size makes it over-fluctuate). When applying catalyst, the catalyst regularizor can suppress the noise from larger step size. After a careful trade off between catalyst regularizor parameter and larger step size, we managed to achieve faster convergence empirically.

**Results.** The results on the two datasets are shown in Figures 2 and 3 respectively. We truncated the #pass and CPU time in subplots (c) and (d) because the F-score has stabilized and we would rather zoom in to see the initial growing phase. In terms of primal gap versus #pass (subplot a), the entropy based method is significantly more effective than Euclidean methods on both datasets (Figure 2(a) and 3(a)). Even with catalyst, Euclidean-Saddle is still much slower than the entropy based methods on the synthetic dataset in Figure 2(a). The CPU time comparisons (subplot b) follow the similar trend, except that the "convex methods" should be ignored because they are introduced only to compare #pass.

The F-score is noisy because, as is well known, it is not monotonic with the primal gap and glitches can appear. In subplots 2(d) and 3(d), the entropy based methods achieve higher F-score significantly faster than the plain Euclidean based methods on both datasets. In terms of passes (subplots 2(c) and 3(c)), Euclidean-Saddle and Entropy-Saddle achieved a similar F-score at first because their primal gaps are comparable at the beginning. After 20 passes, the F-score of Euclidean-Saddle is overtaken by Entropy-Saddle as the primal gap of Entropy-Saddle become much smaller than Euclidean-Saddle.

## 7 Conclusions and Future Work

We have proposed Breg-SVRG to solve saddle-point optimization and proved its linear rate of convergence. Application to adversarial prediction confirmed its effectiveness. For future work, we are interested in relaxing the (potentially hard) proximal update in (21). We will also derive similar reformulations for DCG and precision@$k$, with a quadratic number of variables and with a finite sum structure that is again amenable to Breg-SVRG, leading to a similar reduction of the condition number compared to Euclidean-SVRG. These reformulations, however, come with different constraint sets, and new proximal algorithms with similar complexity as for the F-score can be developed.

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
