[Supplementary Material]

# A  Proofs for Section 2

The following result that sanwiches the Bregman divergence induced by an $L$-smooth convex function is well-known:

**Lemma 2.** *If $f$ is convex, and L-smooth w.r.t. $\|\cdot\|$, then*

$$\tfrac{1}{2L}\left\|\nabla f(x') - \nabla f(x)\right\|_*^2 \leq \Delta_f(x', x) \leq \tfrac{L}{2}\left\|x' - x\right\|^2. \tag{30}$$

*Proof.* The second inequality is obvious. To show the first inequality, define

$$g(x') := \Delta_f(x', x) = f(x') - f(x) - \langle \nabla f(x), x' - x \rangle. \tag{31}$$

Then $g$ is minimized at $x$ with $g(x) = 0$, and $g$ is also $L$-smooth. Therefore for any $\bar{x}$

$$g(\bar{x}) \leq g(x') + \langle \nabla g(x'), \bar{x} - x' \rangle + \frac{L}{2}\left\|\bar{x} - x'\right\|^2 \leq g(x') + \left\|\nabla g(x')\right\|_* \left\|\bar{x} - x'\right\| + \frac{L}{2}\left\|\bar{x} - x'\right\|^2. \tag{32}$$

Now take minimization over $\bar{x}$ on both sides:

$$0 = g(x) \leq g(x') - \frac{1}{2L}\left\|\nabla g(x')\right\|_*^2. \tag{33}$$

Plugging in the definition of $g$ and noticing $\nabla g(x') = \nabla f(x') - \nabla f(x)$, we get the first inequality. □

The following result is crucial for our later analysis, and extends a result of [27].

**Lemma 1.** *Let $f$ and $g$ be $\phi$-saddle and $\varphi$-saddle respectively, with one of them being differentiable. Then, for any $z = (x, y)$ and any saddle point (if exists) $z^* := (x^*, y^*) \in \arg\min_x \max_y \{f(z) + g(z)\}$, we have $f(x, y^*) + g(x, y^*) \geq f(x^*, y) + g(x^*, y) + \Delta_{\phi+\varphi}(z, z^*)$.*

*Proof.* We first recall the following slight generalization of a result of [27]:

**Claim.** Let $h$ and $k$ be respectively $\psi_1$- and $\psi_2$-convex, with one of them being differentiable. Let $x^* \in \arg\min_x h(x) + k(x)$, then for all $x$, $h(x) + k(x) \geq h(x^*) + k(x^*) + \Delta_{\psi_1+\psi_2}(x, x^*)$.

Indeed, using the optimality of $x^*$, we have $\mathbf{0} \in \partial(h+k)(x^*) = \partial h(x^*) + \partial k(x^*)$, where the last equality is due to the differentiable assumption (on one of $h$ and $k$). Since $h$ is $\psi_1$-convex and $k$ is $\psi_2$-convex, we have

$$h(x) \geq h(x^*) + \langle x - x^*, \partial h(x^*) \rangle + \Delta_{\psi_1}(x, x^*) \tag{34}$$
$$k(x) \geq k(x^*) + \langle x - x^*, \partial k(x^*) \rangle + \Delta_{\psi_2}(x, x^*). \tag{35}$$

Adding the above two inequalities and noting that $\Delta_{\psi_1} + \Delta_{\psi_2} = \Delta_{\psi_1+\psi_2}$ completes the proof of our claim.

Now to prove Lemma 1, we note that if $(x^*, y^*)$ is a saddle point of $f + g$, then $x^* \in \arg\min_x f(x, y^*) + g(x, y^*)$ and also $y^* \in \arg\min_y -f(x^*, y) - g(x^*, y)$. Note also that if $f$ is $\phi$-saddle, then $f_y(x) = f(x, y)$ is $\phi_y$-convex. Similarly, if $g$ is $\varphi$-saddle, then $-g_x(y) = -g(x, y)$ is $(-\varphi_x)$-convex. Applying the above claim twice we have:

$$f_{y^*}(x) + g_{y^*}(x) \geq f_{y^*}(x^*) + g_{y^*}(x^*) + \Delta_{\phi_{y^*}+\varphi_{y^*}}(x, x^*) \tag{36}$$
$$-f_{x^*}(y) - g_{x^*}(y) \geq -f_{x^*}(y^*) - g_{x^*}(y^*) + \Delta_{-\phi_{x^*}-\varphi_{x^*}}(y, y^*). \tag{37}$$

Adding the above two equations and noting that $\Delta_{\phi_{y^*}+\varphi_{y^*}}(x, x^*) + \Delta_{-\phi_{x^*}-\varphi_{x^*}}(y, y^*) = \Delta_{\phi+\varphi}(z, z^*)$ completes our proof. □

**Algorithm 2:** SVRG with Bregman Divergence

---

1 Initialize $x_0$ randomly. Set $\tilde{x} = x_0$.
2 **for** $s = 1, 2, \ldots$ **do**                                    // epoch index
3     $\tilde{\mu} \leftarrow \tilde{\mu}^s := \nabla P(\tilde{x}), x_0 \leftarrow x_0^s := x_m^{s-1}$
4     **for** $t = 1, \ldots, m$ **do**                  // iter index
5        Randomly pick $\xi \in \{1, \ldots, n\}$.
6        Compute $v_t$ using (38).
7        Update $x_{t+1}$ using (38).
8     Denote $x_m^s = x_m$.
9     $\tilde{x} \leftarrow \tilde{x}^s := \frac{\sum_{t=1}^m (1+\eta\lambda)^t x_t}{\sum_{t=1}^m (1+\eta\lambda)^t}$.

---

# B   Bregman Divergence for Convex SVRG

Prior to saddle-point optimization, it is illustrative to see how variance reduction methods can be extended to Bregman divergence in convex optimization. Let us consider a proximal objective

$$J(x) = P(x) + \Omega(x) = \tfrac{1}{n} \sum_{k=1}^n \psi_k(x) + \Omega(x).$$

Here each $\psi_k$ is convex and $L$-smooth (w.r.t. some norm), and $\Omega$ is $\Delta$-convex for some Bregman divergence $\Delta$. Breg-SVRG extends the vanilla SVRG by employing the following proximal operator [30] which we assume is efficiently computable:

$$x_{t+1} = \arg\min_x \left\{ \eta \langle v_t, x \rangle + \eta\Omega(x) + \Delta(x, x_t) \right\}, \text{ where } v_t = \nabla\psi_\xi(x_t) - \nabla\psi_\xi(\tilde{x}) + \tilde{\mu}. \quad (38)$$

Here $\xi$ is sampled uniformly at random from $\{1, \ldots, n\}$, $\tilde{x}$ is the pivot found after completing the last epoch, and $\tilde{\mu} = \nabla P(\tilde{x})$. The whole procedure, which we call Breg-SVRG, is detailed in Algorithm 2. To ease notation, the $x_t$ here always refers to the $t$-th step of the current epoch $s$, and we will include the epoch index $s$ only when necessary.

Let us define the gap $\epsilon(x) := J(x) - J(x_*)$ for some $x_*$ that minimizes $J$. Our first convergence result for Algorithm 2 is as follows:

**Theorem 2.** *Assume each $\psi_k$ is convex and $L$-smooth wrt $\|\cdot\|$, and $P$ and $\Omega$ are $(\gamma\Delta)$- and $(\lambda\Delta)$-convex wrt some Bregman divergence $\Delta$, respectively. Let $\eta$ be sufficiently small such that $m := \left\lceil \log(\frac{1}{8\eta L} - \frac{1}{8} - \rho) \big/ \log\rho \right\rceil \geq 1$. Then Breg-SVRG enjoys linear convergence in expectation:*

$$\mathbb{E}\epsilon(\tilde{x}^s) \leq \rho^{-ms}[\Delta(x_*, x_0) + c(Z+1)\epsilon(x_0)],$$

*where $\rho = \frac{1+\eta\lambda}{1-\eta\gamma}, c = \frac{8\eta^2 L}{(1-\eta L)(1-\eta\gamma)}$ and $Z = \sum_{t=0}^{m-1} \rho^t$.*

For example, we may set $\eta = \frac{1}{18L}$, which leads to $c = \frac{4}{153L}$, $m = \frac{\log\left(\frac{9}{8} - \frac{\lambda}{18L}\right)}{\log\left(1 + \frac{\lambda}{18L}\right)} = \Theta\left(\frac{L}{\lambda}\right)$, $(1 + \eta\lambda)^m = \frac{9}{8} - \frac{\lambda}{18L} \geq \frac{9}{8} - \frac{1}{18} = \frac{77}{72}$, and $Z = \frac{9L}{4\lambda} - 1$. Therefore, between epochs, the gap decays by a factor of $\frac{72}{77}$, and each epoch needs to call (38) for $\Theta(L/\lambda)$ times. In total, to reduce the gap below some tolerance $\epsilon$, the proximal operator (38) needs to be called for $O\left(\frac{L}{\lambda} \log\frac{1}{\epsilon}\right)$ times. If the norm $\|\cdot\|$ is chosen to be Euclidean, then the above guarantee reduces to that of SVRG [9]. The condition number $L/\lambda$, however, can change significantly w.r.t. the chosen norm (which reflects the underlying problem geometry).

We need the following variance reduction lemma that extends a result in [9] to any norm.

**Lemma 3.** *The variance of $v_t$ can be bounded by: $\mathbb{E}_\xi \|v_t - \nabla P(x_t)\|_*^2 \leq 16L \cdot [\epsilon(x_t) + \epsilon(\tilde{x})]$.*

*Proof.* Clearly, for any norm, $\|a + b\|^2 \leq 2(\|a\|^2 + \|b\|^2)$. Besides, for any random variable $X$ and norm $\|\cdot\|$, $\mathbb{E}\|X - \mathbb{E}[X]\|^2 \leq 2\mathbb{E}[\|X\|^2 + \|\mathbb{E}X\|^2] \leq 4\mathbb{E}\|X\|^2$. It bounds the "variance" of a random variable, under any norm, by four times its "second moment."

Using these two inequalities and conditional on $x_t$, we have

$$\mathbb{E}_\xi \|v_t - \nabla P(x_t)\|_*^2 = \mathbb{E}_\xi \|(\nabla \psi_\xi(x_t) - \nabla \psi_\xi(\tilde{x})) - (\nabla P(x_t) - \nabla P(\tilde{x}))\|_*^2$$

$$\leq 4 \cdot \mathbb{E}_\xi \|\nabla \psi_\xi(x_t) - \nabla \psi_\xi(\tilde{x})\|_*^2 = 4\mathbb{E}_\xi \|\nabla \psi_\xi(x_t) - \nabla \psi_\xi(x_*) - (\nabla \psi_\xi(\tilde{x}) - \nabla \psi_\xi(x_*))\|_*^2$$

$$\leq 8 \cdot \mathbb{E}_\xi \|\nabla \psi_\xi(x_t) - \nabla \psi_\xi(x_*)\|_*^2 + 8 \cdot \mathbb{E}_\xi \|\nabla \psi_\xi(\tilde{x}) - \nabla \psi_\xi(x_*)\|_*^2. \tag{39}$$

We can next invoke Lemma 2 to upper bound the first part of (39):

$$\frac{1}{2L}\mathbb{E}_\xi \|\nabla \psi_\xi(x_t) - \nabla \psi_\xi(x_*)\|_*^2 \leq \mathbb{E}\Delta_{\psi_\xi}(x_t, x_*) = \Delta_P(x_t, x_*) \leq \epsilon(x_t),$$

where the last inequality is due to (a special case of) Lemma 1. The second part of (39) can be bounded similarly. □

*Proof of Theorem 2.* We apply Lemma 1 to the update (38), with $g = \eta\Omega(x) + \Delta(x, x_t)$, $\phi = 0$, and $\varphi = \lambda\Delta$:

$$\eta \langle v_t, x_{t+1} \rangle + \eta\Omega(x_{t+1}) + \Delta(x_{t+1}, x_t) \tag{40}$$

$$\leq \eta \langle v_t, x^* \rangle + \eta\Omega(x^*) + \Delta(x^*, x_t) - \eta\Delta_\Omega(x^*, x_{t+1}) - \Delta(x^*, x_{t+1}) \tag{41}$$

$$\leq \eta \langle v_t, x^* \rangle + \eta\Omega(x^*) + \Delta(x^*, x_t) - \eta\lambda\Delta(x^*, x_{t+1}) - \Delta(x^*, x_{t+1}). \tag{42}$$

Therefore

$$\eta\Omega(x_{t+1}) + (1 + \eta\lambda)\Delta(x^*, x_{t+1}) \tag{43}$$

$$\leq \Delta(x^*, x_t) + \eta \langle v_t, x^* - x_{t+1} \rangle + \eta\Omega(x^*) - \Delta(x_{t+1}, x_t) \tag{44}$$

$$\leq \Delta(x^*, x_t) + \eta \langle v_t, x^* - x_{t+1} \rangle + \eta\Omega(x^*) - \frac{1}{2}\|x_{t+1} - x_t\|^2, \tag{45}$$

where the last inequality is because $\Delta$ is distance enforcing w.r.t. the norm $\|\cdot\|$. Since $J$ is $L$-smooth, we obtain

$$0 \leq P(x_t) - P(x_{t+1}) + \langle \nabla P(x_t), x_{t+1} - x_t \rangle + \frac{L}{2}\|x_{t+1} - x_t\|^2. \tag{46}$$

Multiplying the above by $\eta > 0$ and adding to (45) we get

$$\eta\Omega(x_{t+1}) + (1 + \eta\lambda)\Delta(x^*, x_{t+1}) \tag{47}$$

$$\leq \Delta(x^*, x_t) + \eta \langle v_t, x^* - x_{t+1} \rangle + \eta\Omega(x^*) - \frac{1 - \eta L}{2}\|x_{t+1} - x_t\|^2 \tag{48}$$

$$+ \eta P(x_t) - \eta P(x_{t+1}) + \eta \langle \nabla P(x_t), x_{t+1} - x_t \rangle \tag{49}$$

$$= \Delta(x^*, x_t) + \eta \langle v_t - \nabla P(x_t), x_t - x_{t+1} \rangle - \frac{1 - \eta L}{2}\|x_{t+1} - x_t\|^2 \tag{50}$$

$$+ \eta P(x_t) - \eta P(x_{t+1}) + \eta\Omega(x^*) + \eta \langle v_t, x^* - x_t \rangle \tag{51}$$

$$\leq \Delta(x^*, x_t) + \frac{\eta^2}{2(1 - \eta L)}\|v_t - \nabla P(x_t)\|_*^2 \tag{52}$$

$$+ \eta[\langle v_t, x^* - x_t \rangle + P(x_t) - P(x_{t+1}) + \Omega(x^*)]. \tag{53}$$

Conditional on $x_t$ we take expectation over $\xi$ on both sides:

$$(1 + \eta\lambda)\mathbb{E}\Delta(x^*, x_{t+1}) \leq (1 - \eta\gamma)\Delta(x^*, x_t) + \frac{\eta^2}{2(1 - \eta L)}\mathbb{E}\|v_t - \nabla P(x_t)\|_*^2 \tag{54}$$

$$+ \eta[J(x^*) - \mathbb{E}J(x_{t+1})], \tag{55}$$

where we have also used the assumption that $P$ is $(\gamma\Delta)$-convex. Using Lemma 3 we take expectation over $x_t$ again on both sides, leading to

$$\rho\Delta_{t+1} \leq \Delta_t + c\left(\delta_t + \tilde{\delta}^{s-1}\right) - \kappa\delta_{t+1}. \tag{56}$$

where $\rho := \frac{1 + \eta\lambda}{1 - \eta\gamma}$, $c := \frac{8\eta^2 L}{(1 - \eta L)(1 - \eta\gamma)}$, $\kappa := \frac{\eta}{1 - \eta\gamma}$, $\delta_t := \mathbb{E}\epsilon(x_t)$, $\Delta_t := \mathbb{E}\Delta(x^*, x_t)$, $\tilde{\delta}^{s-1} := \mathbb{E}\epsilon(\tilde{x}^{s-1})$. Multiplying both sides by $\rho^t$ and telescoping from $t = 0$ to $m - 1$, we obtain

$$\rho^m \Delta_m \leq \Delta_0 + c\sum_{t=1}^{m} \rho^{t-1}\delta_{t-1} + c\tilde{\delta}^{s-1}\sum_{t=1}^{m} \rho^{t-1} - \kappa\sum_{t=1}^{m} \rho^{t-1}\delta_t. \tag{57}$$

Rearranging, we get

$$\rho^m \Delta_m + c\rho^m \delta_m + (\kappa - c\rho) \sum_{t=1}^m \rho^{t-1} \delta_t \le \Delta_0 + c\delta_0 + c\tilde{\delta}^{s-1} \sum_{t=1}^m \rho^{t-1}. \qquad (58)$$

Now define the representer of epoch $s$ as

$$\tilde{x}^s = \frac{1}{Z} \sum_{t=1}^m \rho^{t-1} x_t, \quad \text{where} \quad Z = \sum_{t=1}^m \rho^{t-1}. \qquad (59)$$

Note that $\rho > 1$ hence the most recent iterate gets a bigger weight. Also, we can equivalently use $\tilde{x}^s = \frac{1}{Z'} \sum_{t=1}^m \rho^t x_t$ where $Z' = \sum_{t=1}^m \rho^t$, see Algorithm 2. Then, noting that $J$ is convex and $\tilde{\delta}^s = \mathbb{E}[J(\tilde{x}^s) - J(x^*)]$, we obtain

$$\rho^m(\Delta_m + c\delta_m) + (\kappa - c\rho)Z\tilde{\delta}^s \le \rho^m(\Delta_m + c\delta_m) + (\kappa - c\rho) \sum_{t=1}^m \rho^{t-1} \delta_t \qquad (60)$$

$$\le (\Delta_0 + c\delta_0) + cZ\tilde{\delta}^{s-1}. \qquad (61)$$

Now pick $m$ such that

$$\rho^m = \frac{(\kappa - c\rho)Z}{cZ} = \frac{\kappa - c\rho}{c} = \frac{1 - \eta L}{8\eta L} - \frac{1 + \eta\lambda}{1 - \eta\gamma} = \frac{1}{8\eta L} - \frac{1}{8} - \frac{1 + \eta\lambda}{1 - \eta\gamma}. \qquad (62)$$

Therefore,

$$\mathbb{E}\Delta(x^*, x_m^s) + c(\mathbb{E}J(x_m^s) - J(x^*)) + cZ(J(\tilde{x}^s) - J(x^*)) \qquad (63)$$

$$\le \rho^{-m}[\mathbb{E}\Delta(x^*, x_m^{s-1}) + c(\mathbb{E}J(x_m^{s-1}) - J(x^*)) + cZ(J(\tilde{x}^{s-1}) - J(x^*))]. \qquad (64)$$

So there is a decay of factor $\rho^{-m}$ between epochs. Set $\eta = \frac{\alpha}{L}$ and we obtain

$$\rho^m = \frac{1}{8\alpha} - \frac{1}{8} - \frac{L + \alpha\lambda}{L - \alpha\gamma} > 1 \qquad (65)$$

for $\alpha$ sufficiently small. Moreover, $\rho = \frac{L + \alpha\lambda}{L - \alpha\gamma}$ hence

$$m = \frac{\log\left(\frac{1}{8\alpha} - \frac{1}{8} - \frac{L+\alpha\lambda}{L-\alpha\gamma}\right)}{\log \frac{L+\alpha\lambda}{L-\alpha\gamma}} = \Theta\left(\frac{1}{\alpha} \log \frac{1}{\alpha} \cdot \frac{L}{\lambda + \gamma}\right). \qquad (66)$$

So between epochs, we decrease the gap by a constant factor that is strictly smaller than 1, and the number of iterations per epoch is $\Theta\left(\frac{L}{\lambda+\gamma}\right)$. In total, to find an $\epsilon$ accurate solution, the computational cost is $\Theta\left(\frac{L}{\lambda+\gamma} \log \frac{1}{\epsilon}\right)$. $\qquad \square$

## C  Rates for Proximal Saddle-Point Optimization in Section 4

The proof of [17] relies on resolvent operators, which is inherently restricted to the Euclidean norm. Besides, their bound is on $\|z_t - z^*\|^2$, and it was claimed that "the convex minimization analysis does not apply and we use the notion of monotone operators to prove convergence". We show here that by introducing an auxiliary variable, our analysis in Appendix B can be largely reused for SVRG with Bregman divergence in saddle-point problems, and the bound is directly on function values.

**Theorem 1.** *Let Assumption 1 hold, and choose a sufficiently small $\eta > 0$ such that $m := \left\lceil \log\left(\frac{1-\eta L}{18\eta L^2} - \eta - 1\right)/\log(1+\eta)\right\rceil \ge 1$. Then Breg-SVRG enjoys linear convergence in expectation:*

$$\mathbb{E}\epsilon(\tilde{z}^s) \le (1+\eta)^{-ms}[\Delta(z^*, z_0) + c(Z+1)\epsilon(z_0)], \text{ where } Z = \sum_{t=0}^{m-1}(1+\eta)^t, \ c = \frac{18\eta^2 L^2}{1-\eta L}. \quad (22)$$

*Proof.* Our key innovation in analysis is the introduction of an auxiliary variable: $u_t = \begin{pmatrix} \partial_x K(x_t, y^*) \\ -\partial_y K(x^*, y_t) \end{pmatrix}$. Note that $\mathbb{E}_\xi v_t \ne u_t$.

Recall that

$$\epsilon_t^M := \epsilon^M(z_t) := M(x_t, y^*) - M(x^*, y_t), \qquad \epsilon_t^K := \epsilon^K(z_t) := K(x_t, y^*) - K(x^*, y_t) \quad (67)$$

$$\epsilon_t^K := \epsilon^K(z_t) := K(x_t, y^*) - K(x^*, y_t) \in \mathbb{R} \tag{68}$$

$$\epsilon_t = \epsilon(z_t) = J(x_t, y^*) - J(x^*, y_t) = [J(x_t, y^*) - J(x^*, y^*)] + [J(x^*, y^*) - J(x^*, y_t)] \quad (69)$$

$$\geq \Delta(z_t, z^*) \geq \tfrac{1}{2} \|z_t - z^*\|^2 \geq 0. \tag{70}$$

The first step of our proof is to invoke Lemma 1 on the update (21):

$$(1+\eta)\Delta(z^*, z_{t+1}) \leq \Delta(z^*, z_t) - \eta\epsilon_{t+1}^M - \Delta(z_{t+1}, z_t) + \eta\langle v_t, z^* - z_t \rangle + \eta\langle v_t, z_t - z_{t+1}\rangle$$

$$= \Delta(z^*, z_t) - \eta\epsilon_{t+1}^M - \Delta(z_{t+1}, z_t) + \eta\langle v_t, z^* - z_t\rangle + \eta\langle v_t - u_t, z_t - z_{t+1}\rangle + \eta\langle u_t, z_t - z_{t+1}\rangle.$$

It is easy to bound $\langle u_t, z_t - z_{t+1}\rangle$ as $K$ is $L$-smooth:

$$\langle u_t, z_t - z_{t+1}\rangle = \langle \partial_x K(x_t, y^*), x_t - x_{t+1}\rangle - \langle \partial_y K(x^*, y_t), y_t - y_{t+1}\rangle$$

$$\leq K(x_t, y^*) - K(x_{t+1}, y^*) + \tfrac{L}{2}\|x_t - x_{t+1}\|^2 + K(x^*, y_{t+1}) - K(x^*, y_t) + \tfrac{L}{2}\|y_t - y_{t+1}\|^2$$

$$= \epsilon_t^K - \epsilon_{t+1}^K + \tfrac{L}{2}\|z_t - z_{t+1}\|^2. \tag{71}$$

So we can proceed by

$$(1+\eta)\Delta(z^*, z_{t+1})$$

$$\leq \Delta(z^*, z_t) - \eta\epsilon_{t+1} + \eta\epsilon_t^K + \eta\langle v_t, z^* - z_t\rangle + \eta\langle v_t - u_t, z_t - z_{t+1}\rangle - \tfrac{1-\eta L}{2}\|z_t - z_{t+1}\|^2$$

$$\leq \Delta(z^*, z_t) - \eta\epsilon_{t+1} + \eta\epsilon_t^K + \eta\langle v_t, z^* - z_t\rangle + \frac{\eta^2\|v_t - u_t\|_*^2}{2(1-\eta L)}.$$

Take expectation over $\xi$ on both sides (conditional on $z_t$). Since $\mathbb{E}_\xi[v_t] = \mathsf{G}(z_t)$, we may apply the inequality $K(x, y') - K(x', y) \leq \langle \mathsf{G}(z), z - z'\rangle$:

$$(1+\eta)\mathbb{E}\Delta(z^*, z_{t+1}) \leq \Delta(z^*, z_t) - \eta\mathbb{E}\epsilon_{t+1} + \frac{\eta^2}{2(1-\eta L)}\mathbb{E}\|v_t - u_t\|_*^2. \tag{72}$$

Finally we bound $\mathbb{E}\|v_t - u_t\|_*^2$:

$$\mathbb{E}\|v_t - u_t\|_*^2 = \mathbb{E}\|v_t - \mathsf{G}(z_t) + \mathsf{G}(z_t) - u_t\|_*^2 \leq 2\mathbb{E}\|v_t - \mathsf{G}(z_t)\|_*^2 + 2\|\mathsf{G}(z_t) - u_t\|_*^2. \tag{73}$$

Notice that by the $L$-smoothness of $K$,

$$\|\mathsf{G}(z_t) - u_t\|_*^2 = \|\partial_x K(x_t, y_t) - \partial_x K(x_t, y^*)\|_*^2 + \|\partial_y K(x_t, y_t) - \partial_y K(x^*, y_t)\|_*^2$$

$$\leq L^2\|y_t - y^*\|^2 + L^2\|x_t - x^*\|^2. \tag{74}$$

Again using $\mathbb{E}\|X - \mathbb{E}[X]\|^2 \leq 4\mathbb{E}\|X\|^2$ and $L$-smoothness of $\psi_k$,

$$\mathbb{E}\|v_t - \mathsf{G}(z_t)\|_*^2 = \mathbb{E}\|\nabla\psi_\xi(z_t) - \nabla\psi_\xi(\tilde{z}) - \mathbb{E}[\nabla\psi_\xi(z_t) - \nabla\psi_\xi(\tilde{z})]\|_*^2$$

$$\leq 4\mathbb{E}\|\nabla\psi_\xi(z_t) - \nabla\psi_\xi(\tilde{z})\|_*^2 \leq 4L^2\|z_t - \tilde{z}\|^2$$

$$\leq 8L^2\|z_t - z^*\|^2 + 8L^2\|\tilde{z} - z^*\|^2. \tag{75}$$

Plug (74) and (75) into (73), and then into (72). Using (67), we finally arrive at (expectation on $\xi$)

$$(1+\eta)\mathbb{E}\Delta(z^*, z_{t+1}) \leq \Delta(z^*, z_t) - \eta\mathbb{E}\epsilon_{t+1} + \frac{18\eta^2 L^2}{1-\eta L}(\epsilon_t + \epsilon(\tilde{z}^{s-1})). \tag{76}$$

Taking expectation of the whole history on both sides, we obtain

$$\rho\Delta_{t+1} \leq \Delta_t + c'\left(\delta_t + \tilde{\delta}^{s-1}\right) - \eta\delta_{t+1}. \tag{77}$$

where $\rho := 1 + \eta$, $c' := \frac{18\eta^2 L^2}{1-\eta L}$, $\delta_t := \mathbb{E}\epsilon(z_t)$, $\Delta_t := \mathbb{E}\Delta(z^*, z_t)$, $\tilde{\delta}^{s-1} := \mathbb{E}\epsilon(\tilde{z}^{s-1})$. This has exactly the same shape as (56), and therefore the rest derivation is almost identical, except that in $c'$,

we have $\eta^2 L^2$ rather that $\eta^2 L$ as under (56). So almost all the derivation can be shared. Let us set $\eta = \frac{1}{45L^2}$, and we obtain

$$\rho^m = \frac{\eta - c\rho}{c} = \frac{45 - 1/L}{18} - \frac{1}{45L^2} - 1 \geq \frac{45 - 1}{18} - \frac{1}{45} - 1 = \frac{64}{45}. \tag{78}$$

Since $\rho = 1 + \frac{1}{45L^2}$, we derive

$$m = \frac{\log\left(\frac{45 - 1/L}{18} - \frac{1}{45L^2} - 1\right)}{\log\left(1 + \frac{1}{45L^2}\right)} = \Theta\left(L^2\right). \tag{79}$$

So between epochs, the decay is by a factor of $\frac{45}{64}$, and the number of iterations per epoch is $\Theta(L^2)$. The total computational cost is therefore $O\left(L^2 \log \frac{1}{\epsilon}\right)$. $\qquad\square$

## D    Efficient Proximal Operator for Solving (27)

Given a set of variables $\{\boldsymbol{\alpha}_i\}$ which are *not* necessarily in $S$, we need to project it to $S$ based on Bregman divergence. Here we show how this can be done in $O(n^2)$ time for both Euclidean and entropic projections.

### D.1    Euclidean projection to $S$

Given a set $\{\boldsymbol{\alpha}_k\}$, the projection to $S$ requires solving

$$\min_{\{\mathbf{x}_k\}} \tfrac{1}{2}\sum_k \|\mathbf{x}_k - \boldsymbol{\alpha}_k\|_2^2 \text{ s.t. } \mathbf{x}_k \in C_k, \ \sum_k \mathbf{1}'\mathbf{x}_k \leq 1 \tag{80}$$

$$\text{where} \quad C_k := \{\mathbf{x} \in [0, \tfrac{r_k}{k}]^n : r_k \geq 0, \ \mathbf{1}'\mathbf{x} = r_k\} \tag{81}$$

Introducing a Lagrange variable $\rho$ that corresponds to the last constraint, we obtain the partial Lagrangian:

$$\max_{\rho \geq 0} -\rho + \sum_k \min_{\mathbf{x}_k \in C_k} \left\{\frac{1}{2}\|\mathbf{x}_k - \boldsymbol{\alpha}_k\|_2^2 + \rho\mathbf{1}'\mathbf{x}_k\right\}. \tag{82}$$

Since the optimal $\mathbf{x}_k$ is unique by strong convexity, we can solve $\rho$ by any smooth solver such as BFGS, proximal bundle method (PBM, http://napsu.karmitsa.fi/proxbundle/), or even bi-section. Given a $\rho$ and its optimal $\mathbf{x}_k(\rho)$, the gradient in $\rho$ can be easily computed as $-1 + \sum_k \mathbf{1}'\mathbf{x}_k(\rho)$. Therefore it suffices to optimize $\mathbf{x}_k$ separately. In the sequel, we will first present the optimization procedure without worrying about the computational cost. After that, we will show how to reduce the complexity to $\tilde{O}(n)$.

Fixing $\rho$, the optimal $\mathbf{x}_k$ can be found by solving the following problem. Here we dropped all subscripts $k$ to lighten the notation.

$$\min_{r \geq 0} \min_{\mathbf{x} \in [0, \frac{r}{k}]^n, \ \mathbf{1}'\mathbf{x}=r} \frac{1}{2}\|\mathbf{x} - \boldsymbol{\alpha}\|_2^2 + \rho\mathbf{1}'\mathbf{x}. \tag{83}$$

Introducing a Lagrange multiplier $\mu$ for the $\mathbf{1}'\mathbf{x} = r$ constraint, we dualize the inner problem as

$$\min_{r \geq 0} \min_{\mathbf{x} \in [0, \frac{r}{k}]^n} \max_{\mu} \frac{1}{2}\|\mathbf{x} - \boldsymbol{\alpha}\|_2^2 + \rho\mathbf{1}'\mathbf{x} - \mu(\mathbf{1}'\mathbf{x} - r)$$

$$= \min_{r \geq 0} \max_{\mu} \left\{\mu r + \min_{\mathbf{x} \in [0, \frac{r}{k}]^n} \frac{1}{2}\|\mathbf{x} - \boldsymbol{\alpha}\|_2^2 + \rho\mathbf{1}'\mathbf{x} - \mu\mathbf{1}'\mathbf{x}\right\} \tag{84}$$

$$= \min_{r \geq 0} \max_{\mu} \left\{\mu r + \min_{\mathbf{y} \in [0,1]^n} \frac{1}{2}\left\|\tfrac{r}{k}\mathbf{y} - \boldsymbol{\alpha}\right\|_2^2 + (\rho - \mu)\tfrac{r}{k}\mathbf{1}'\mathbf{y}\right\}. \tag{85}$$

Now $r$ does not appear in the constraint and so the once we have the optimal $\mu$ and $\mathbf{y}$ (or optimal $\mathbf{x}$ based on which we get the optimal $\mathbf{y} = \frac{k}{r}\mathbf{x}$), the gradient in $r$ can be written as

$$\mu + \mathbf{y}'(\tfrac{r}{k}\mathbf{y} - \boldsymbol{\alpha})/k + (\rho - \mu)\mathbf{1}'\mathbf{y}/k. \tag{86}$$

which can be calculated in constant time via our efficient update rule.

| **Algorithm 3:** Euclidean projection of $\{\boldsymbol{\alpha}_k\}$ on $S$ | **Algorithm 4:** Entropic projection of $\{\boldsymbol{\alpha}_k\}$ on $S$ |
|---|---|
| **1** $\rho^* = \text{minimize}(\texttt{obj\_rho}, [0, +\infty))$ | **1** $\rho^* = \text{minimize}(\texttt{obj\_rho}, [0, +\infty))$ |
| **2** $[\sim, \sim, \{\mathbf{x}_k\}] = \texttt{obj\_rho}(\rho^*)$. | **2** $[\sim, \sim, \{\mathbf{x}_k\}] = \texttt{obj\_rho}(\rho^*)$. |
| **3 Return** $\{\mathbf{x}_k\}$ | **3 Return** $\{\mathbf{x}_k\}$ |
| **Function** $[f, g, \{\mathbf{x}_k\}] = \texttt{obj\_rho}(\rho)$ | **Function** $[f, g, \{\mathbf{x}_k\}] = \texttt{obj\_rho}(\rho)$ |
| **4 for** $k = 1, \dots, n$ **do** | **4 for** $k = 1, \dots, n$ **do** |
| **5** $\quad r_k = \text{minimize}(\texttt{@(r)obj\_r}(r, k, \rho),$ $\quad\quad [0, +\infty))$ | **5** $\quad r_k = \text{minimize}(\texttt{@(r)obj\_r}(r, k, \rho),$ $\quad\quad [0, +\infty))$ |
| **6** $\quad [f_k, \sim, \mathbf{x}_k] = \texttt{obj\_r}(r_k, k, \rho)$ $\quad\quad\quad\quad \triangleright \mathbf{x}_k = \mathbf{x}_k(r_k) = \mathbf{x}_k(\rho)$ | **6** $\quad [f_k, \sim, \mathbf{x}_k] = \texttt{obj\_r}(r_k, k, \rho)$ $\quad\quad\quad\quad \triangleright \mathbf{x}_k = \mathbf{x}_k(r_k) = \mathbf{x}_k(\rho)$ |
| **7** $f = \rho - \sum_{k=1}^n f_k$ | **7** $f = \rho - \sum_{k=1}^n f_k$ |
| **8** $g = 1 - \sum_{k=1}^n \mathbf{1}'\mathbf{x}_k$ | **8** $g = 1 - \sum_{k=1}^n \mathbf{1}'\mathbf{x}_k$ |
| **end function** | **end function** |
| **Function** $[f, g, \mathbf{x}] = \texttt{obj\_r}(r, k, \rho)$ | **Function** $[f, g, \mathbf{x}] = \texttt{obj\_r}(r, k, \rho)$ |
| **9** $\mu_{\min} = \rho - \max_s \alpha_{ks},$ $\quad \mu_{\max} = \rho - \min_s \alpha_{ks} + \frac{r}{k}$ | **9** $\mu_{\min} = -50,$ $\quad \mu_{\max} = \rho + \log(\frac{r}{k * \min_s \alpha_{ks}})$ |
| **10 while** *true* **do** $\quad\quad \triangleright$ bi-section search | **10 while** *true* **do** $\quad\quad \triangleright$ bi-section search |
| **11** $\quad \mu = (\mu_{\min} + \mu_{\max})/2$ | **11** $\quad \mu = (\mu_{\min} + \mu_{\max})/2$ |
| **12** $\quad \mathbf{x} = \text{MED}(\boldsymbol{\alpha}_k + \mu\mathbf{1} - \rho\mathbf{1}, \mathbf{0}, \frac{r}{k}\mathbf{1})$ | **12** $\quad \mathbf{x} = \text{MIN}(\boldsymbol{\alpha}\exp(\mu - \rho), \frac{r}{k}\mathbf{1})$ |
| **13** $\quad$ **if** $\mathbf{1}'\mathbf{x} > r + 10^{-5}$ **then** | **13** $\quad$ **if** $\mathbf{1}'\mathbf{x} > r + 10^{-5}$ **then** |
| **14** $\quad\quad \mu_{\max} = \mu$ | **14** $\quad\quad \mu_{\max} = \mu$ |
| **15** $\quad$ **else if** $\mathbf{1}'\mathbf{x} < r - 10^{-5}$ **then** | **15** $\quad$ **else if** $\mathbf{1}'\mathbf{x} < r - 10^{-5}$ **then** |
| **16** $\quad\quad \mu_{\min} = \mu$ | **16** $\quad\quad \mu_{\min} = \mu$ |
| **17** $\quad$ **else** | **17** $\quad$ **else** |
| **18** $\quad\quad$ **break** | **18** $\quad\quad$ **break** |
| **19** $f = \frac{1}{2}\|\mathbf{x} - \boldsymbol{\alpha}_k\|_2^2 + \rho\mathbf{1}'\mathbf{x} \; \triangleright$ Now $\mathbf{x} = \mathbf{x}_k(r)$ | **19** $f = \rho\mathbf{1}'\mathbf{x} + \sum_s Q \quad \triangleright$ Now $\mathbf{x} = \mathbf{x}_k(r)$ |
| **20** $g = \mu + \mathbf{y}'(\frac{r}{k}\mathbf{y} - \boldsymbol{\alpha})/k + (\rho - \mu)\mathbf{1}'\mathbf{y}/k \quad \triangleright$ Now $\mu = \mu_k(r)$ | **20** $g = \mu + \sum_s \frac{y_s}{k}\log\frac{y_s r}{\alpha_s k} + (\rho - \mu)\frac{y_s}{k} \quad \triangleright$ Now $\mu = \mu_k(r)$ |
| **end function** | **end function** |

Given $r$ and $\mu$, the optimal $\mathbf{x}$ admits a closed form

$$\mathbf{x} = \text{MED}(\boldsymbol{\alpha} + \mu\mathbf{1} - \rho\mathbf{1}, \mathbf{0}, \tfrac{r}{k}\mathbf{1}), \tag{87}$$

where MED stands for the elementwise median. Given $r$, the optimal $\mu$ is the one that ensures $\mathbf{1}'\mathbf{x} = r$ (*not* necessarily unique). Since each $x$ in (87) is non-decreasing in $\mu$, a simple bi-section search can find such a $\mu(r)$ by probing $O(\log n)$ values of $\mu$. With $\mu(r)$ in hand, the optimal $\mathbf{x}(r)$ for the inner problem in (83) can be recovered by (87).

In hindsight, we observe that although the optimal $\mathbf{x}$ in (83) is unique, the objective function in $r$ is not necessarily smooth because $r$ also appears in the constraints of $\mathbf{x}$. Therefore we resort to a nonsmooth solver (e.g. PBM) for optimizing over $r$.

The overall procedure for Euclidean projection is summarized in Algorithm 3. We assumed without loss of generality that for each $\boldsymbol{\alpha}_k$ all its elements are already sorted increasingly. The binary search over $\mu$ can be refined, with $\mu$ only probing kink points corresponding to the entries in $\boldsymbol{\alpha}_k$. This will ensure the bi-section terminates in $O(\log\min\{n, 1/\epsilon\})$ iterations.

## D.2 Entropic projection to $S$

Given a set $\{\boldsymbol{\alpha}_k\}$, the entropic projection to $S$ requires solving

$$\min_{\{\mathbf{x}_k\}} \sum_{ks} x_{ks}\log\frac{x_{ks}}{\alpha_{ks}} + \alpha_{ks} - x_{ks} \tag{88}$$

$$s.t. \; \mathbf{x}_k \in C_k, \; \sum_{ks} x_{ks} \leq 1 \quad \text{where} \quad C_k := \{\mathbf{x} \in [0, \tfrac{r_k}{k}]^n : r_k \geq 0, \; \mathbf{1}'\mathbf{x} = r_k\}$$

Introducing a Lagrange variable $\rho$ that corresponds to the last constraint, we obtain the partial Lagrangian:

$$\max_{\rho \geq 0} -\rho + \sum_{ks} \min_{\mathbf{x}_k \in C_k} \left\{ x_{ks} \log \frac{x_{ks}}{\alpha_{ks}} + \alpha_{ks} - x_{ks} + \rho x_{ks} \right\}. \tag{89}$$

Since the optimal $\mathbf{x}_k$ is unique by strong convexity, we can solve $\rho$ by any smooth solver such as BFGS, PBM, or even bi-section. Given a $\rho$ and its optimal $\mathbf{x}_k(\rho)$, the gradient in $\rho$ can be easily computed as $-1 + \sum_k \mathbf{1}'\mathbf{x}_k(\rho)$. Therefore it suffices to optimize $\mathbf{x}_k$ separately.

Fixing $\rho$, the optimal $\mathbf{x}_k$ can be found by solving the following problem. Here we dropped all subscripts $k$ to lighten the notation.

$$\min_{r \geq 0} \min_{\mathbf{x} \in [0, \frac{r}{k}]^n, \, \mathbf{1}'\mathbf{x}=r} \sum_s \{Q + \rho x_s\} \quad \text{where} \quad Q = x_s \log \frac{x_s}{\alpha_s} + \alpha_s - x_s \tag{90}$$

Introducing a Lagrange multiplier $\mu$ for the $\mathbf{1}'\mathbf{x} = r$ constraint, we dualize the inner problem as

$$\min_{r \geq 0} \max_{\mu} \left\{ \mu r + \min_{x_s \in [0, \frac{r}{k}]} \sum_s \{Q + \rho x_s - \mu x_s\} \right\} \tag{91}$$

$$= \min_{r \geq 0} \max_{\mu} \left\{ \mu r + \min_{y_s \in [0,1]} \sum_s \{Q_y + (\rho - \mu)\frac{r}{k}y_s\} \right\} \tag{92}$$

$$\text{where } Q_y = \frac{r}{k}y_s \log \frac{y_s r}{\alpha_s k} + \alpha_s - \frac{r}{k}y_s$$

the gradient in $r$ can be written as

$$\mu + \sum_s \frac{y_s}{k} \log \frac{y_s r}{\alpha_s k} + (\rho - \mu)\frac{y_s}{k}. \tag{93}$$

which can be calculated in constant time via our efficient update rule.

Given $r$ and $\mu$, the optimal $\mathbf{x}$ admits a closed form

$$\mathbf{x} = \text{MIN}\left(\boldsymbol{\alpha} \exp(\mu - \rho), \frac{r}{k}\mathbf{1}\right), \tag{94}$$

where MIN stands for the elementwise minimum. Given $r$, the optimal $\mu$ is the one that ensures $\mathbf{1}'\mathbf{x} = r$ (*not* necessarily unique). Since each $x$ in (94) is non-decreasing in $\mu$, a simple bi-section search can find such a $\mu(r)$ by probing $O(\log n)$ values of $\mu$. With $\mu(r)$ in hand, the optimal $\mathbf{x}(r)$ for the inner problem in (90) can be recovered by (94).

The overall procedure for Entropic projection is summarized in Algorithm 4. We assumed without loss of generality that for each $\boldsymbol{\alpha}_k$ all its elements are already sorted increasingly. The binary search over $\mu$ can be refined, with $\mu$ only probing kink points corresponding to the entries in $\boldsymbol{\alpha}_k$. This will ensure the bi-section terminates in $O(\log n) < O(\log 1/\epsilon)$ iterations.

# E Rates for Proximal Saddle-Point Optimization (Non-uniform)

**Problem.** We consider the following problem

$$(x^*, y^*) = \arg\min_x \max_y K(x, y) + M(x, y), \quad \text{where} \quad K(x, y) = \frac{1}{n}\sum_{i=1}^n \psi_i(x, y). \tag{95}$$

**Assumption 2.** We assume each $\psi_i$ is a saddle function that is $L_i$-smooth as follows:

$$L_i = \sup_{z \neq z'} \frac{\|B_i(z) - B_i(z')\|_*}{\|z - z'\|}, \quad \text{where} \quad B_i(z) = [\partial_x \psi_i(x, y); -\partial_y \psi_i(x, y)] \tag{96}$$

and $K$ is $L_{avg}$-smooth:

$$L_{avg} = \sup_{z \neq z'} \frac{\|B(z) - B(z')\|_*}{\|z - z'\|}, \quad \text{where} \quad B(z) = [\partial_x \psi(x, y); -\partial_y \psi(x, y)] \tag{97}$$

Then we can define $\bar{L}$ adapted to our sampling schemes:

$$\bar{L}(\pi)^2 = \sup_{z \neq z'} \frac{\sum_{i=1}^n \frac{1}{n^2 \pi_i} \|B_i(z) - B_i(z')\|_*^2}{\|z - z'\|^2}, \quad \text{where} \quad B_i(z) = [\partial_x \psi_i(x, y); -\partial_y \psi_i(x, y)] \tag{98}$$

and $\pi$ is a probability vector that sums to 1. We always have the bound:

$$L_{avg}^2 \leq \bar{L}(\pi)^2 \leq \max_{i=1}^n L_i^2 \times \sum_{i=1}^n \frac{1}{n^2 \pi_i}. \tag{99}$$

**Algorithm.** Let us define a variant of variance-reduced stochastic gradient for saddle-point problems:

$$v_t := [v_x(z_t); -v_y(z_t)], \tag{100}$$

$$\text{where} \quad v_x(z_t) := \frac{1}{n \pi_\xi}(\partial_x \psi_\xi(z_t) - \partial_x \psi_\xi(\tilde{z})) + \partial_x K(\tilde{z}) \tag{101}$$

$$v_y(z_t) := \frac{1}{n \pi_\xi}(\partial_y \psi_\xi(z_t) - \partial_y \psi_\xi(\tilde{z})) + \partial_y K(\tilde{z}). \tag{102}$$

Here $\tilde{z}$ is the pivot chosen after completing the last epoch, $\xi$ is randomly choose from probability vector $\pi$. Clearly, $\mathbb{E}_\xi[v_t] = \mathsf{G}(z_t)$ (unbiased). The stochastic algorithm then performs the proximal update at each step:

$$(x_{t+1}, y_{t+1}) = \arg\min_x \max_y \eta \langle v_x(z_t), x \rangle + \eta \langle v_y(z_t), y \rangle + \eta M(x, y) + \Delta(x, x_t) - \Delta(y, y_t). \tag{103}$$

**Theorem 3.** *With the above modification, the same guarantee in Theorem 1 with $L$ (in fact, this $L$ is $\max_{i=1}^n L_i$) replaced by $\bar{L} = \bar{L}(\pi)$ holds.*

*Proof.* Our key innovation in analysis is the introduction of an auxiliary variable:

$$u_t = \begin{pmatrix} \partial_x K(x_t, y^*) \\ -\partial_y K(x^*, y_t) \end{pmatrix}. \tag{104}$$

Note that $\mathbb{E}_\xi v_t \neq u_t$. The first step of our proof is to invoke Lemma 1 on the update (103):

$$(1 + \eta)\Delta(z^*, z_{t+1}) \leq \Delta(z^*, z_t) - \eta \epsilon_{t+1}^M - \Delta(z_{t+1}, z_t) + \eta \langle v_t, z - z_t \rangle + \eta \langle v_t, z_t - z_{t+1} \rangle \tag{105}$$

$$= \Delta(z^*, z_t) - \eta \epsilon_{t+1}^M - \Delta(z_{t+1}, z_t) + \eta \langle v_t, z - z_t \rangle \tag{106}$$
$$+ \eta \langle v_t - u_t, z_t - z_{t+1} \rangle + \eta \langle u_t, z_t - z_{t+1} \rangle.$$

It is easy to bound $\langle u_t, z_t - z_{t+1} \rangle$ as $K$ is $L_{avg}$-smooth:

$$\langle u_t, z_t - z_{t+1} \rangle = \langle \partial_x K(x_t, y^*), x_t - x_{t+1} \rangle - \langle \partial_y K(x^*, y_t), y_t - y_{t+1} \rangle \tag{107}$$
$$\leq K(x_t, y^*) - K(x_{t+1}, y^*) + \tfrac{L_{avg}}{2} \|x_t - x_{t+1}\|^2$$
$$+ K(x^*, y_{t+1}) - K(x^*, y_t) + \tfrac{L_{avg}}{2} \|y_t - y_{t+1}\|^2 \tag{108}$$
$$= \epsilon_t^K - \epsilon_{t+1}^K + \tfrac{L_{avg}}{2} \|z_t - z_{t+1}\|^2 \tag{109}$$
$$= \epsilon_t^K - \epsilon_{t+1}^K + \tfrac{\bar{L}}{2} \|z_t - z_{t+1}\|^2. \tag{110}$$

The last inequality is due to $L_{avg}^2 \leq \bar{L}^2$.

So we can proceed by

$$(1 + \eta)\Delta(z^*, z_{t+1}) \leq \Delta(z^*, z_t) - \eta \epsilon_{t+1} + \eta \epsilon_t^K + \eta \langle v_t, z^* - z_t \rangle + \eta \langle v_t - u_t, z_t - z_{t+1} \rangle \tag{111}$$
$$- \tfrac{1 - \eta \bar{L}}{2} \|z_t - z_{t+1}\|^2$$

$$\leq \Delta(z^*, z_t) - \eta \epsilon_{t+1} + \eta \epsilon_t^K + \eta \langle v_t, z^* - z_t \rangle + \frac{\eta^2 \|v_t - u_t\|_*^2}{2(1 - \eta \bar{L})}. \tag{112}$$

Take expectation over $\xi$ on both sides (conditional on $z_t$). Since $\mathbb{E}_\xi[v_t] = \mathsf{G}(z_t)$, we apply the inequality $K(x, y') - K(x', y) \le \langle \mathsf{G}(z), z - z' \rangle$ and get:

$$(1 + \eta)\mathbb{E}\Delta(z^*, z_{t+1}) \le \Delta(z^*, z_t) - \eta\mathbb{E}\epsilon_{t+1} + \frac{\eta^2}{2(1 - \eta\bar{L})}\mathbb{E}\|v_t - u_t\|_*^2. \tag{113}$$

Finally we bound $\mathbb{E}\|v_t - u_t\|_*^2$:

$$\begin{aligned}
\mathbb{E}\|v_t - u_t\|_*^2 &= \mathbb{E}\|v_t - \mathsf{G}(z_t) + \mathsf{G}(z_t) - u_t\|_*^2 \\
&\le 2\mathbb{E}\|v_t - \mathsf{G}(z_t)\|_*^2 + 2\|\mathsf{G}(z_t) - u_t\|_*^2.
\end{aligned} \tag{114}$$

Notice that by the $L$-smoothness of $K$,

$$\|\mathsf{G}(z_t) - u_t\|_*^2 = \|\partial_x K(x_t, y_t) - \partial_x K(x_t, y^*)\|_*^2 + \|\partial_y K(x_t, y_t) - \partial_y K(x^*, y_t)\|_*^2 \tag{115}$$
$$\le L_{avg}^2\|y_t - y^*\|^2 + L_{avg}^2\|x_t - x^*\|^2 \tag{116}$$
$$\le \bar{L}^2\|y_t - y^*\|^2 + \bar{L}^2\|x_t - x^*\|^2. \tag{117}$$

Again using the definition (98),

$$\begin{aligned}
\mathbb{E}\|v_t - \mathsf{G}(z_t)\|_*^2 &= \mathbb{E}\|\frac{1}{n\pi_\xi}(\nabla\psi_\xi(z_t) - \nabla\psi_\xi(\tilde{z})) - \mathbb{E}[\frac{1}{n\pi_\xi}(\nabla\psi_\xi(z_t) - \nabla\psi_\xi(\tilde{z}))]\|_*^2 \\
&\le 4\mathbb{E}\left\|\frac{1}{n\pi_\xi}(\nabla\psi_\xi(z_t) - \nabla\psi_\xi(\tilde{z}))\right\|_*^2 \\
&\le 4\bar{L}^2\|z_t - \tilde{z}\|^2 \quad \text{(by } \bar{L}\text{-smoothness)} \\
&\le 8\bar{L}^2\|z_t - z^*\|^2 + 8\bar{L}^2\|\tilde{z} - z^*\|^2.
\end{aligned} \tag{118}$$

Plug (116) and (118) into (114), and then into (113). Using (67), we finally arrive at (expectation is only over $\xi$)

$$(1 + \eta)\mathbb{E}\Delta(z^*, z_{t+1}) \le \Delta(z^*, z_t) - \eta\mathbb{E}\epsilon_{t+1} + \frac{18\bar{L}^2\eta^2}{1 - \eta\bar{L}}(\epsilon_t + \epsilon(\tilde{z}^{s-1})). \tag{119}$$

Taking expectation of the whole history on both sides, we obtain

$$\rho\Delta_{t+1} \le \Delta_t + c'\left(\delta_t + \tilde{\delta}^{s-1}\right) - \eta\delta_{t+1}. \tag{120}$$

where $\rho := 1 + \eta$, $c' := \frac{18\bar{L}^2\eta^2}{1-\eta\bar{L}}$, $\delta_t := \mathbb{E}\epsilon(z_t)$, $\Delta_t := \mathbb{E}\Delta(z^*, z_t)$, $\tilde{\delta}^{s-1} := \mathbb{E}\epsilon(\tilde{z}^{s-1})$. This has exactly the same shape as (56), and therefore the rest derivation is almost identical, except that in $c'$, we have $\bar{L}^2\eta^2$ rather that $\eta^2 L_Q$ as under (56). So almost all the derivation can be shared. Let us set $\eta = \frac{1}{45\bar{L}^2}$, and we obtain

$$\rho^m = \frac{\eta - c\rho}{c} = \frac{45 - 1/\bar{L}}{18} - \frac{1}{45\bar{L}^2} - 1 \ge \frac{45 - 1}{18} - \frac{1}{45} - 1 = \frac{64}{45}. \tag{121}$$

Since $\rho = 1 + \frac{1}{45\bar{L}^2}$, we derive

$$m = \frac{\log\left(\frac{45 - 1/\bar{L}}{18} - \frac{1}{45\bar{L}^2} - 1\right)}{\log\left(1 + \frac{1}{45\bar{L}^2}\right)} = \Theta\left(\bar{L}^2\right). \tag{122}$$

So between epochs, the decay is by a factor of $\frac{45}{64}$, and the number of iterations per epoch is $\Theta(\bar{L}^2)$. The total computational cost is therefore $O\left(\bar{L}^2\log\frac{1}{\epsilon}\right)$.

For uniform sampling, $\pi_i = 1/n$ for all $i = 1, \ldots, n$, we can recover Theorem 1 as $\bar{L} = \max_i L_i$ in this case. The smallest possible value for $\bar{L}$ is $\bar{L} = L_{avg} = (1/n)\sum_{i=1}^n L_i$, achieved at $\pi_i = L_i/\sum_{j=1}^n L_j$, i.e., the sampling probabilities for the component functions are proportional to their Lipschitz constants.

$\square$

## F  More about $L$

Let us consider both $\boldsymbol{\alpha}$ and $\boldsymbol{\beta}$ as $n^2$ dimensional vectors. Denote $\mathbf{z} = \begin{pmatrix} \boldsymbol{\alpha} \\ \boldsymbol{\beta} \end{pmatrix}$. Then the bilinear part

of $\frac{1}{n^2} \sum_{ij} f_{ij}(\boldsymbol{\alpha}_i, \boldsymbol{\beta}_j)$ can be written as $F(\boldsymbol{\alpha}, \boldsymbol{\beta}) = \frac{1}{2}\boldsymbol{\alpha}'A\boldsymbol{\beta} = \frac{1}{2}\mathbf{z}' \begin{pmatrix} \mathbf{0} & A \\ A' & \mathbf{0} \end{pmatrix} \mathbf{z}$. Then $\nabla F(\mathbf{z}) = \begin{pmatrix} \mathbf{0} & A \\ A' & \mathbf{0} \end{pmatrix} \mathbf{z} = \begin{pmatrix} A\boldsymbol{\beta} \\ A'\boldsymbol{\alpha} \end{pmatrix}$

Recall that $\|\mathbf{z}\|^2 = \|\boldsymbol{\alpha}\|^2 + \|\boldsymbol{\beta}\|^2$ and similarly for their dual norms. We could use subscripts for these norms to highlight that $\boldsymbol{\alpha}$, $\boldsymbol{\beta}$, and $\mathbf{z}$ employ different norms. But we omit these subscripts because they are clear from the context.

So the $L^2$ of $F$ can be computed as

$$L^2 = \max_{\|\mathbf{z}\| \leq 1} \left\| \begin{pmatrix} A\boldsymbol{\beta} \\ A'\boldsymbol{\alpha} \end{pmatrix} \right\|_*^2 = \max_{\|\boldsymbol{\alpha}\|^2 + \|\boldsymbol{\beta}\|^2 \leq 1} \|A\boldsymbol{\beta}\|_*^2 + \|A'\boldsymbol{\alpha}\|_*^2. \tag{123}$$

The objective function here is obviously convex in $(\boldsymbol{\alpha}, \boldsymbol{\beta})$ jointly. Since we are maximizing, the optimal solution must be attained at some extreme points of the domain. There can be only two types of extreme points: a) $\|\boldsymbol{\alpha}\| = 1$ and $\boldsymbol{\beta} = \mathbf{0}$; and b) $\boldsymbol{\alpha} = \mathbf{0}$ and $\|\boldsymbol{\beta}\| = 1$. So

$$L^2 = \max \left\{ \max_{\|\boldsymbol{\alpha}\|=1} \|A'\boldsymbol{\alpha}\|_*^2, \ \max_{\|\boldsymbol{\beta}\|=1} \|A\boldsymbol{\beta}\|_*^2 \right\}, \tag{124}$$

where the first term corresponds to case a), and the second term to case b). It is not hard to see from definition that these two terms are equal, both being exactly $\max_{\|\boldsymbol{\alpha}\|=\|\boldsymbol{\beta}\|=1} \boldsymbol{\alpha}'A\boldsymbol{\beta}$.

Now let us add the quadratic term in $\boldsymbol{\alpha}$, so that $G(\boldsymbol{\alpha}, \boldsymbol{\beta}) = \frac{1}{2}\boldsymbol{\alpha}'A\boldsymbol{\beta} + \frac{1}{2}\boldsymbol{\alpha}'B\boldsymbol{\alpha}$. Then

$$\nabla G(\mathbf{z}) = \begin{pmatrix} B & A \\ A' & \mathbf{0} \end{pmatrix} \mathbf{z} = \begin{pmatrix} B\boldsymbol{\alpha} + A\boldsymbol{\beta} \\ A'\boldsymbol{\alpha} \end{pmatrix} \tag{125}$$

So we can compute the $L$ of $G$ by:

$$L^2 = \max_{\|\mathbf{z}\| \leq 1} \left\| \begin{pmatrix} B\boldsymbol{\alpha} + A\boldsymbol{\beta} \\ A'\boldsymbol{\alpha} \end{pmatrix} \right\|_*^2 = \max_{\|\boldsymbol{\alpha}\|^2 + \|\boldsymbol{\beta}\|^2 \leq 1} \|B\boldsymbol{\alpha} + A\boldsymbol{\beta}\|_*^2 + \|A'\boldsymbol{\alpha}\|_*^2 \tag{126}$$

$$\leq \max \left\{ \max_{\|\boldsymbol{\alpha}\|=1} \|B\boldsymbol{\alpha}\|_*^2 + \|A'\boldsymbol{\alpha}\|_*^2, \ \max_{\|\boldsymbol{\beta}\|=1} \|A\boldsymbol{\beta}\|_*^2 \right\} \tag{127}$$

$$\leq \max_{\|\boldsymbol{\alpha}\|=1} \|B\boldsymbol{\alpha}\|_*^2 + \max_{\|\boldsymbol{\alpha}\|=1} \|A'\boldsymbol{\alpha}\|_*^2. \tag{128}$$

where the last equality again used the fact that $\max_{\|\boldsymbol{\alpha}\|=1} \|A'\boldsymbol{\alpha}\|_*^2 = \max_{\|\boldsymbol{\beta}\|=1} \|A\boldsymbol{\beta}\|_*^2$, and $\max_{\boldsymbol{\alpha}}\{f(\boldsymbol{\alpha}) + g(\boldsymbol{\alpha})\} \leq \max_{\boldsymbol{\alpha}} f(\boldsymbol{\alpha}) + \max_{\boldsymbol{\alpha}} g(\boldsymbol{\alpha})$.

So now bounding $\max_{\|\boldsymbol{\alpha}\|=1} \|B\boldsymbol{\alpha}\|_*^2$ can be done in the similar way as bounding $\max_{\|\boldsymbol{\alpha}\|=1} \|A'\boldsymbol{\alpha}\|_*^2$.

## G  Stochastic Updates of SVRG on F-score game

The original problem for F-score is

$$\min_{\{\boldsymbol{\alpha}_i\} \in S} \max_{\{\boldsymbol{\beta}_j\} \in S} \left\{ \frac{1}{n^2} \sum_{i=1}^{n} \sum_{j=1}^{n} f_{ij}(\boldsymbol{\alpha}_i, \boldsymbol{\beta}_j) + \max_{\boldsymbol{\theta}} -\frac{\lambda}{2} \|\boldsymbol{\theta}\|_2^2 + \frac{\boldsymbol{\theta}'}{n} X\tilde{\mathbf{y}} - \frac{\boldsymbol{\theta}'}{n} X \sum_{i=1}^{n} i\boldsymbol{\alpha}_i \right\}. \tag{129}$$

Then the optimal $\boldsymbol{\theta}$ admits a closed form solution

$$\boldsymbol{\theta}(\boldsymbol{\alpha}) = \frac{1}{\lambda n} X \left( \tilde{\mathbf{y}} - \sum_{i=1}^{n} i\boldsymbol{\alpha}_i \right). \tag{130}$$

Plugging it back and denoting $\mathbf{c} = X\tilde{\mathbf{y}}$, we arrive at the overall problem in $\boldsymbol{\alpha}$ and $\boldsymbol{\beta}$ only:

$$\min_{\{\boldsymbol{\alpha}_i\} \in S} \max_{\{\boldsymbol{\beta}_j\} \in S} \frac{1}{n^2} \sum_{i=1}^{n} \sum_{j=1}^{n} \left[ f_{ij}(\boldsymbol{\alpha}_i, \boldsymbol{\beta}_j) - \frac{i}{\lambda n} \mathbf{c}' X \boldsymbol{\alpha}_i + \frac{ij}{2\lambda} \boldsymbol{\alpha}_i' X' X \boldsymbol{\alpha}_j + \frac{1}{2\lambda n^2} \|\mathbf{c}\|_2^2 \right]. \quad (131)$$

$$= \min_{\{\boldsymbol{\alpha}_i\} \in S} \max_{\{\boldsymbol{\beta}_j\} \in S} \frac{1}{n} \sum_{j=1}^{n} \psi_j(\boldsymbol{\alpha}, \boldsymbol{\beta}_j). \quad (132)$$

where

$$\psi_j(\boldsymbol{\alpha}, \boldsymbol{\beta}_j) = \frac{1}{n} \sum_{i=1}^{n} \left[ f_{ij}(\boldsymbol{\alpha}_i, \boldsymbol{\beta}_j) - \frac{i}{\lambda n} \mathbf{c}' X \boldsymbol{\alpha}_i + \frac{ij}{2\lambda} \boldsymbol{\alpha}_i' X' X \boldsymbol{\alpha}_j + \frac{1}{2\lambda n^2} \|\mathbf{c}\|_2^2 \right]$$

Denoting $\mathbf{e}_j = (0, 0, \ldots, 1, \ldots, 0)$, i.e. the canonical row vector for $j$th dimension. Thus,

1. The stochastic gradient over $\boldsymbol{\alpha}^t$ at iteration $t$ is

$$\nabla \psi_j(\boldsymbol{\alpha}^t, \boldsymbol{\beta}_j^t)$$

$$= \nabla \frac{1}{n} \sum_{i=1}^{n} \left[ f_{ij}(\boldsymbol{\alpha}_i^t, \boldsymbol{\beta}_j^t) - \frac{i}{\lambda n} \mathbf{c}' X \boldsymbol{\alpha}_i^t + \frac{ij}{2\lambda} \boldsymbol{\alpha}_i^{t'} X' X \boldsymbol{\alpha}_j^t + \frac{1}{2\lambda n^2} \|\mathbf{c}\|_2^2 \right]$$

$$= \nabla \left[ \frac{1}{n} \sum_{i=1}^{n} f_{ij}(\boldsymbol{\alpha}_i^t, \boldsymbol{\beta}_j^t) \right] - \frac{X'\mathbf{c} \cdot (1:n)}{\lambda n^2} + \frac{jX'X[\boldsymbol{\alpha}_j^t \cdot (1:n) + (\sum_{i=1}^{n} i\boldsymbol{\alpha}_i^t)\mathbf{e}_j]}{2\lambda n}$$

2. The delayed stochastic gradient over anchor variable $\hat{\boldsymbol{\alpha}}$ is

$$\nabla \psi_j(\hat{\boldsymbol{\alpha}}, \hat{\boldsymbol{\beta}}_j)$$

$$= \nabla \left[ \frac{1}{n} \sum_{i=1}^{n} f_{ij}(\hat{\boldsymbol{\alpha}}_i, \hat{\boldsymbol{\beta}}_j) \right] - \frac{X'\mathbf{c} \cdot (1:n)}{\lambda n^2} + \frac{jX'X[\hat{\boldsymbol{\alpha}}_j \cdot (1:n) + (\sum_{i=1}^{n} i\hat{\boldsymbol{\alpha}}_i)\mathbf{e}_j]}{2\lambda n}$$

3. Full gradient in each epoch is

$$\hat{\mu} = \frac{1}{n} \sum_{j=1}^{n} \nabla \psi_j(\hat{\boldsymbol{\alpha}}, \hat{\boldsymbol{\beta}}_j)$$

$$= \nabla \left[ \frac{1}{n^2} \sum_{i=1}^{n} \sum_{j=1}^{n} f_{ij}(\hat{\boldsymbol{\alpha}}_i, \hat{\boldsymbol{\beta}}_j) \right] - \frac{X'\mathbf{c} \cdot (1:n)}{\lambda n^2} + \frac{X'X(\sum_{j=1}^{n} j\hat{\boldsymbol{\alpha}}_j) \cdot (1:n)}{\lambda n^2}$$

$$= \nabla \left[ \frac{1}{n^2} \sum_{i=1}^{n} \sum_{j=1}^{n} f_{ij}(\hat{\boldsymbol{\alpha}}_i, \hat{\boldsymbol{\beta}}_j) \right] - X'\boldsymbol{\theta}(\hat{\boldsymbol{\alpha}}) * \frac{(1:n)}{n}$$

So the euclidean SVRG update in each iteration is

$$\boldsymbol{\alpha}^{t+1} = \boldsymbol{\alpha}^t - \eta \left[ \hat{\mu} + \nabla \psi_j(\boldsymbol{\alpha}^t, \boldsymbol{\beta}_j^t) - \nabla \psi_j(\hat{\boldsymbol{\alpha}}, \hat{\boldsymbol{\beta}}_j) \right] \quad (133)$$