[Reviews · NeurIPS 2017]

Reviewer 1



This paper shows that "certain" adversarial prediction problems under multivariate losses can be solved "much faster than they used to be". The paper stands on two main ideas: (1) that the general saddle function optimization problem stated in eq. (9) can be simplified from an exponential to a quadratic complexity (on the sample size), and (2) that the simplified optimization problem, with some regularization, can be solved using some extension of the SVRG (stochastic variance reduction gradient) method to Bregman divergences. The paper is quite focused on the idea of obtaining a faster solution of the adversarial problem. However, the key simplification is applied to a specific loss, the F-score, so one may wonder if the benefits of the proposed method could be extended to other losses. The extension of the SVRG is a more general result, it seems that the paper could have been focused on proposing Breg-SVRG, showing the adversarial optimization with the F-score as a particular application. In any case, I think the paper is technically correct and interesting enough to be accepted.

Reviewer 2



I found this an interesting paper with strong technical results (although it is far from my areas of expertise, do I don't have high confidence in my review). A few comments: * Sections 3 and 4 feel a little bit disjointed - they are two very separate threads, and it feels like the paper lacks a little focus. Would it make sense to put section 4 first? * The inclusion of the regularizer ||\theta||^2 in (10) is a little abrupt - not very well explained - more explanation/justification would be helpful. * The application to LP boosting using entropy regularization would benefit from more discussion: how does the resulting algorithm compare to the methods introduced in the original paper?

Reviewer 3



The paper extends a saddle point derivative of the SVRG algorithm to the case of entropy regularization (rather than Euclidean). The algorithm is applied to the problem of adversarial prediction, where the authors simplify the objective function so that it can be solved efficiently. The algorithm is then evaluated experimentally. The only comment that I have about the paper is its presentation and writing. It seems the authors have tried to cram a lot of material into such a short paper. This makes the paper hard-to-follow at times and its reading tiresome. Nonetheless, the contributions are novel albeit straight-forward, and as such I suggest its acceptance. References ======== P. Balamurugan and F. Bach. Stochastic variance reduction methods for saddle-point problems.